# The plant mobile domain proteins MAIN and MAIL1 interact with the phosphatase PP7L to regulate gene expression and silence transposable elements in *Arabidopsis thaliana*

Melody Nicolau[1,2], Nathalie Picault[1,2], Julie Descombin[1,2], Yasaman Jami-Alahmadi[3], Suhua Feng[4], Etienne Bucher[5], Steven E. Jacobsen[4,6], Jean-Marc Deragon[1,2,7], James Wohlschlegel[3], Guillaume Moissiard[1,2]*

1 LGDP-UMR5096, CNRS, Perpignan, France, 2 LGDP-UMR5096, Université de Perpignan, France, 3 Department of Biological Chemistry, University of California at Los Angeles, Los Angeles, California, United States of America, 4 Department of Molecular, Cell and Developmental Biology, University of California at Los Angeles, Los Angeles, California, United States of America, 5 Plant Breeding and Genetic Resources, Agroscope, Nyon, Switzerland, 6 Howard Hughes Medical Institute, University of California at Los Angeles, Los Angeles, California, United States of America, 7 Institut Universitaire de France, Paris, France

* guillaume.moissiard@univ-perp.fr

**Data Availability Statement:** Nucleotide sequencing data generated in this study have been

## Abstract

Transposable elements (TEs) are DNA repeats that must remain silenced to ensure cell integrity. Several epigenetic pathways including DNA methylation and histone modifications are involved in the silencing of TEs, and in the regulation of gene expression. In *Arabidopsis thaliana*, the TE-derived plant mobile domain (PMD) proteins have been involved in TE silencing, genome stability, and control of developmental processes. Using a forward genetic screen, we found that the PMD protein MAINTENANCE OF MERISTEMS (MAIN) acts synergistically and redundantly with DNA methylation to silence TEs. We found that MAIN and its close homolog MAIN-LIKE 1 (MAIL1) interact together, as well as with the phosphoprotein phosphatase (PPP) PP7-like (PP7L). Remarkably, *main*, *mail1*, *pp7l* single and *mail1 pp7l* double mutants display similar developmental phenotypes, and share common subsets of upregulated TEs and misregulated genes. Finally, phylogenetic analyses of PMD and PP7-type PPP domains among the Eudicot lineage suggest neo-association processes between the two protein domains to potentially generate new protein function. We propose that, through this interaction, the PMD and PPP domains may constitute a functional protein module required for the proper expression of a common set of genes, and for silencing of TEs.

## Author summary

The plant mobile domain (PMD) is a protein domain of unknown function that is widely spread in the angiosperm plants. Although most PMDs are associated with repeated DNA sequences called transposable elements (TEs), plants have domesticated the PMD to

deposited in European Nucleotide Archive (ENA) under the accession number PRJEB33240 (http://www.ebi.ac.uk/ena/data/view/PRJEB33240). The proteomics data have been deposited to the MassIVE data repository (https://massive.ucsd.edu) with the dataset identifier MSV000084089.

**Funding:** This study was supported by the French Laboratory of Excellence project "TULIP" (ANR-10-LABX-41; ANR-11-IDEX-0002-02), by a UPVD BQR grant, and by a grant from the French CNRS GDR EPIPLANT consortium. Work in the Jacobsen laboratory was supported by NIH grant 1R35GM130272 and a grant from the W.M. Keck Foundation. S. E. J. is an Investigator of the Howard Hughes Medical Center. E.B. was supported by ERC 725701 BUNGEE. The funders had no role in study design, data collection and analysis, decision to publish, or preparation of the manuscript.

**Competing interests:** The authors have declared that no competing interests exist.

produce genic versions that play important roles within the cell. In *Arabidopsis thaliana*, MAINTENANCE OF MERISTEMS (MAIN) and MAIN-LIKE 1 (MAIL1) are genic PMDs that are involved in genome stability, developmental processes, and silencing of TEs. The mechanisms involving MAIN and MAIL1 in these cellular processes remain elusive. Here, we show that MAIN, MAIL1 and the phosphoprotein phosphatase (PPP) named PP7-like (PP7L) interact to form a protein complex that is required for the proper expression of genes, and the silencing of TEs. Phylogenetic analyses revealed that PMD and PP7-type PPP domains are evolutionary connected, and several plant species express proteins carrying both PMD and PPP domains. We propose that interaction of PMD and PPP domains would create a functional protein module involved in mechanisms regulating gene expression and repressing TEs.

## Introduction

In eukaryotes, DNA methylation and post-translational modifications of histones are epigenetic marks involved in chromatin organization, regulation of gene expression and silencing of DNA repeats such as transposable elements (TEs) [1–3]. Constitutive heterochromatin is highly condensed and enriched in silenced TEs that are targeted by DNA methylation and histone H3 lysine 9 dimethylation (H3K9me2). Euchromatin is more relaxed and composed of genes that are more permissive to transcription, depending on the recruitment of transcription factors (TFs), cofactors and RNA polymerases [1, 4]. In plants, DNA methylation occurs in three different cytosine contexts: CG, CHG and CHH (where H = A, T or C), involving specialized DNA methyltransferases [5]. In *Arabidopsis thaliana*, DOMAINS REARRANGED METHYLTRANSFERASE 2 (DRM2) and DRM1 mediate de novo DNA methylation in all sequence contexts through the RNA-directed DNA methylation (RdDM) pathway, which involves among other components, RNA-DEPENDENT RNA POLYMERASE 2 (RDR2) and DICER-LIKE 3 (DCL3) for the production of short interfering (si)RNAs [6, 7]. The maintenance of CG methylation is specifically performed by METHYLTRANSFERASE 1 (MET1), while CHROMOMETHYLASE 2 (CMT2) and CMT3 are involved in the maintenance at CHG sites [8, 9]. CMT2 can also be involved in the deposition of CHH methylation at specific genomic location [10, 11]. Finally, DRM2 is mostly required for the maintenance of CHH methylation through the RdDM pathway [6, 7, 9]. Together with DNA methylation, additional pathways play important roles in TE silencing. The MICRORCHIDIA 1 (MORC1) and MORC6 ATPases interact together, and are required for heterochromatin condensation and repression of TEs, acting mostly downstream of DNA methylation and RdDM pathway [12–14].

More recently, the *A. thaliana* plant mobile domain (PMD) proteins MAINTENANCE OF MERISTEM (MAIN) and MAIN-LIKE 1 (MAIL1) were identified as new factors required for TE silencing [15]. In addition, these two proteins have been involved in genome stability, and regulation of developmental processes such as cell division and differentiation [16, 17]. The PMD is a large protein domain of unknown function that is widely represented among the angiosperms, predominantly associated with TEs [15, 18]. It has been proposed that genic PMD versions, such as the MAIN and MAIL1 proteins derived from TEs after gene domestication [15, 18, 19]. Previous studies suggested that genic PMDs could act as cellular factors related to transcription, possibly acting as transcription factor (TF)-like, co-factor or repressor proteins regulating this cellular process [16, 18]. Nevertheless, the role of PMD proteins in the regulation of transcription remains elusive. Most of genic PMD proteins are standalone

versions, however, in some cases, the PMD is fused to another protein domain, such as protease, kinase or metallo-phosphatase (MPP) domains. For instance in *A. thaliana*, the MAIL3 protein carries a PMD, which is fused to a putative serine/threonine-specific phosphoprotein phosphatase (PPP) domain phylogenetically related to the plant-specific protein phosphatase 7 (PP7) [20]. PP7 is a calmodulin-binding PPP that has been related to cryptochrome (CRY)-mediated blue-light signaling, and to the control of stomatal aperture [20–22]. PP7 is also involved in the perception of red/far red light by controlling the phytochrome pathway [23, 24]. In addition to PP7 and MAIL3 (also known as "long PP7"), the protein PP7-like (PP7L) belongs to the same phylogenetic clade [20]. PP7L was recently identified as a nuclear protein involved in chloroplast development and abiotic stress tolerance [25]. The *pp7l* mutant plants showed photosynthetic defects and strong developmental phenotype associated with misregulation of several genes [25].

In this study, we described a forward genetic screen based on a GFP reporter gene that allowed us to identify a mutant population in which *MAIN* is mutated, leading to GFP overexpression. We then deciphered the genetic interaction between the DRM2, CMT3 and MAIN, showing that these proteins are part of different epigenetic pathways that act redundantly or synergistically to repress TEs. Biochemical analyses indicated that MAIN and MAIL1 physically interact together. These analyses also identified PP7L as a robust interactor of MAIN and MAIL1 proteins. In addition, the characterization of developmental and molecular phenotypes of *pmd* and *pp7l* single and double mutant plants strongly suggest that these proteins interact together to silence TEs, and regulate the expression of a common set of genes. Finally, phylogenetic analyses allowed us to determine the distribution of PMD and PP7/PP7L domains among the Eudicots. Based on these analyses, we have evidences of co-evolution linked to the neo-association of the PMD and PP7-type PPP domains on single proteins in several Eudicot species, suggesting a convergent evolution between these two protein domains.

## Results

### Mutation in *MAIN* is responsible for TE silencing defects

The *ATCOPIA28* retrotransposon *AT3TE51900* (hereafter called *ATCOPIA28*) is targeted by several epigenetic pathways such as DNA methylation and the MORC1/6 complex, which altogether contribute to its repression. We engineered a construct in which the 5' long terminal repeat (LTR) promoter region of *ATCOPIA28* controls GFP transcription (Fig 1A). While the *ATCOPIA28*::*GFP* transgene is fully silenced in wild type (WT) plants, it is weakly expressed in the DNA methylation-deficient *drm1 drm2 cmt3 (ddc)* triple mutant background (Fig 1B) [26]. We performed an ethyl methane sulfonate (EMS) mutagenesis using the *ATCOPIA28*::*GFP ddc* plants as sensitized genetic material, and screened for mutant populations showing GFP overexpression. Among, the selected populations, we retrieved two new mutant alleles of *MORC6* carrying missense mutations in either the GHKL or S5 domains of the protein (S1A–S1C Fig). We also identified the population *ddc #16* showing strong overexpression of GFP and misregulation of several endogenous TEs, including *ATCOPIA28* (Fig 1B–1D). Mapping experiments based on whole genome resequencing and bulk segregant analysis indicated that *ddc #16* carries a missense point mutation (C230Y) in the gene *AT1G17930*, previously named *MAIN* (S1D and S1E Fig). Genetic complementation analyses by crossing the *ddc #16* EMS mutant with the knock-out (KO) transferred DNA (T-DNA) insertion line *main-2* generated F1 *ddc #16* x *main-2* plants that did not express the GFP (S1F Fig). Transcriptional profiling analyses showed, however, that endogenous TEs, including *ATCOPIA28*, were upregulated in F1 *ddc #16* x *main-2* plants, but not in F1 control plants generated from the backcross of *ddc #16* with WT Columbia (Col) plants (S1G Fig). Self-fertilization of F1 *ddc #16* x *main-2* plants

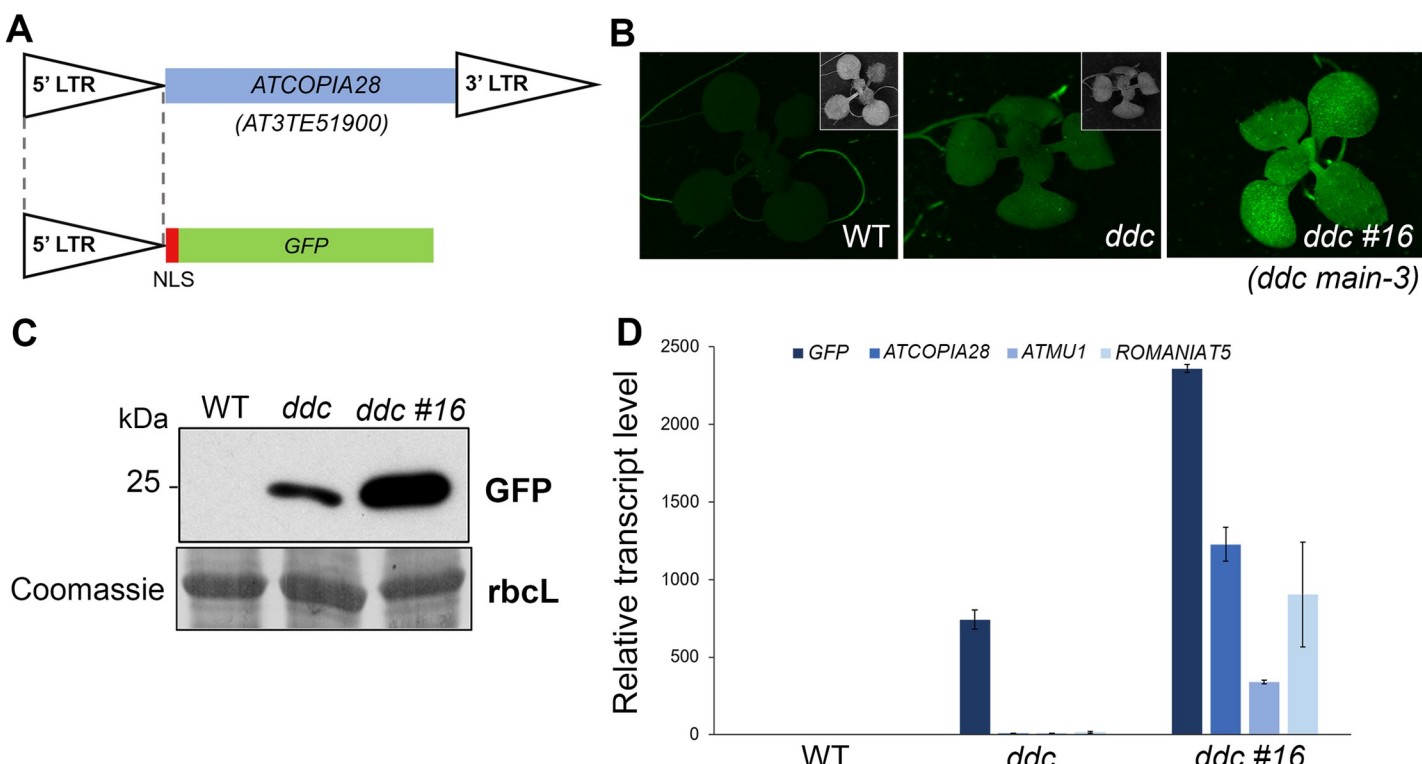

**Fig 1. The *ddc #16* EMS population shows overexpression of *ATCOPIA28::GFP* and upregulation of endogenous TEs.** (A) Schematic representation of the *ATCOPIA28::GFP* transgene. The 5' long terminal repeat (LTR) promoter region of an *ATCOPIA28* LTR-retrotransposon (*AT3TE51900*) is used to control the expression of GFP. The construct carries a Nuclear Localization Signal (NLS) to target the GFP in the nucleus. (B) WT and *drm1 drm2 cmt3* (*ddc*) triple mutant plants carrying the *ATCOPIA28*::GFP transgene showed no and weak GFP fluorescence under UV light, respectively. By comparison, the *ddc #16* EMS mutant showed strong GFP fluorescence. Insets show plants under white light. (C) Western blot using anti-GFP antibody confirmed *ATCOPIA28::GFP* overexpression in *ddc #16*. Coomassie staining of the large Rubisco subunit (rbcL) is used as a loading control. KDa: kilodalton. (D) Relative expression analyses of *ATCOPIA28::GFP (GFP)* and three endogenous TEs in *ddc* and *ddc #16* assayed by Real-Time quantitative PCR (RT-qPCR). RT-qPCR analyses were normalized using the housekeeping *RHIP1* gene, and transcript levels in the mutants are represented relative to WT. Error bars indicate standard deviation based on three independent biological replicates. Screening of EMS mutant populations was done on MS plates to allow for visualization of GFP-positive individuals under UV light.

allowed us to retrieve several F2 *ddc #16* x *main-2* plants overexpressing the GFP (S1F Fig). Among these GFP positive F2 plants, we identified individuals that were either homozygote for the EMS mutation in the *MAIN* gene, or plants carrying both the EMS and T-DNA *main-2* mutant alleles (S1F Fig). Moreover, while all these plants were homozygote for the *drm2* mutation, half of them segregated the *cmt3* mutation. Thus, altogether, these analyses suggested that *ATCOPIA28*::*GFP* silencing is more DRM2- than CMT3-dependent. More importantly, they confirmed that *MAIN* was the mutated gene causing the upregulation of *ATCOPIA28*::*GFP* and several endogenous TEs. Therefore, *ddc #16* was renamed *ddc main-3*.

## The MAIN, DRM2 and CMT3 pathways act synergistically to repress TEs and DNA-methylated genes

To determine the genetic interaction of *ddc* and *main-3* mutations on TE silencing, we carried out two independent RNA sequencing (RNA-seq) experiments in the hypomorphic *main-3* single, *ddc* triple and *ddc main-3* quadruple mutant plants (Fig 2A and S2A Fig). As previously described, the *ddc* mutant showed upregulation of several TEs spread over the five chromosomes (Fig 2B–2D and S2B Fig and S1 Table) [11]. Loss of TE silencing was also observed to a milder degree in the *main-3* mutant, with the significant enrichment of pericentromeric TEs

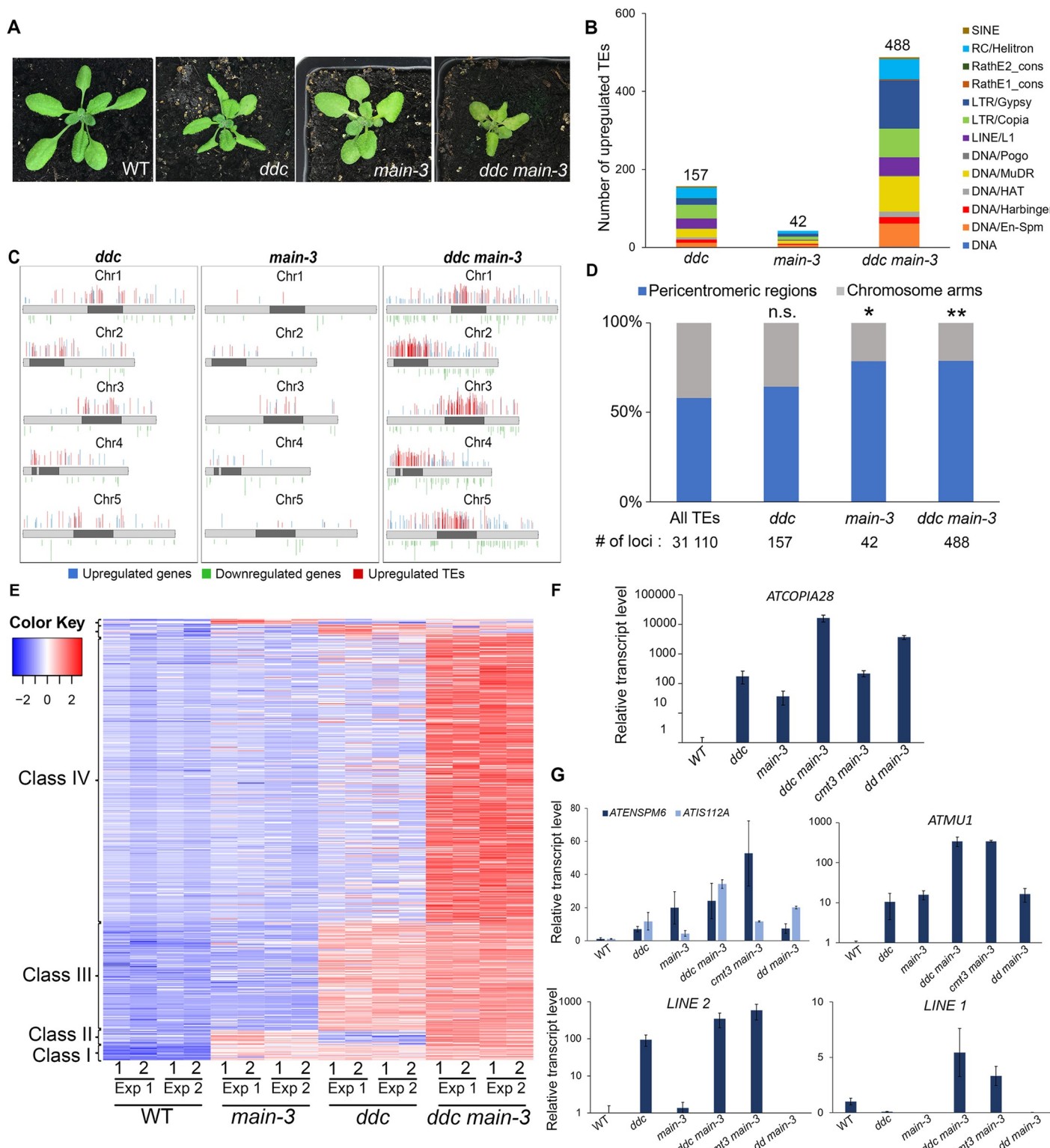

**Fig 2. MAIN, DRM2 and CMT3 act synergistically to repress TEs.** (A) Representative pictures showing the developmental phenotype of 3-week-old *ddc*, *main-3* and *ddc main-3* mutants in comparison to WT plant. (B) Number of upregulated TEs in *ddc*, *main-3* and *ddc main-3*, and classified by TE superfamily. (C) Chromosomal distributions of misregulated loci in *ddc*, *main-3* and *ddc main-3* over WT. Chromosome arms are depicted in light grey, pericentromeric regions in dark grey as defined in [50]. Upregulated genes and TEs are represented in blue and red, respectively; downregulated genes are represented in green. (D) Fraction of upregulated TEs in *ddc*, *main-3* and *ddc main-3* located in chromosome arms or in pericentromeric regions as defined in [50]. Asterisks indicate statistically

significant enrichments of TEs in pericentromeric regions in comparison to the genomic distribution of all *A. thaliana* TEs (Chi-Square test, *: p-value≤ 0.05, **: p-value≤ 0.01 n.s: not significant). (E) Heatmap showing upregulated TEs in *ddc*, *main-3* and *ddc main-3* mutants in comparison to WT plants. (F-G) Relative expression analyses of *ATCOPIA28* (F) and several endogenous TEs (G) in *ddc*, *main-3*, *ddc main-3*, *cmt3 main-3* and *drm1 drm2 (dd) main-3* assayed by RT-qPCR. RT-qPCR analyses were normalized using the housekeeping *RHIP1* gene, and transcript levels in the mutants are represented relative to WT. Error bars indicate standard deviation based on three independent biological replicates. RNA-seq threshold: log2≥2, or log2≤-2; p-adj< 0.01.

among the upregulates TEs (Fig 2B–2D and S2B Fig and S1 Table). The *ddc main-3* mutant showed an exacerbation of TE silencing defects, with a large number of pericentromeric TEs being specifically upregulated in this mutant background (Fig 2B–2D and S2B Fig and S1 Table). Comparative analyses revealed that upregulated TEs cluster into four distinct classes (Fig 2E and S2C Fig). Class I TEs are upregulated in *ddc*, *main-3* and *ddc main-3* mutants (Fig 2E and S2C and S2D Fig). Class II and class III TEs are targeted by the MAIN and DRM2/CMT3 pathways, respectively (Fig 2E and S2C and S2D Fig). However, the upregulation of class II and class III TEs is further enhanced in *ddc main-3*, which suggests that the MAIN and DRM2/CMT3 pathways can partially compensate each other at these genomic locations (S2D Fig). Finally, the most abundant class IV TEs are only misregulated in *ddc main-3*, which implies that the MAIN and DRM2/CMT3 pathways act redundantly to silence these TEs (Fig 2E and S2C and S2D Fig).

Several genes were also misregulated in the three mutant backgrounds (S1 Table). Among these genes, a subset was commonly upregulated in *ddc*, *main-3* and *ddc main-3* (S2E Fig). Remarkably, genes that were upregulated in *ddc*, *main-3* or *ddc main-3* were significantly enriched in pericentromeric regions of chromosomes, where constitutive heterochromatin resides (S2F Fig). This is consistent with the fact that, among these upregulated genes, we identified a large proportion of genes that were DNA-methylated (in the three cytosine contexts) and targeted by H3K9me2 (S2F Fig). Conversely, we could only identify one gene commonly downregulated in *ddc*, *main-3* and *ddc main-3* (S2F Fig). Furthermore, downregulated genes in *ddc*, *main-3* or *ddc main-3* were rather enriched in chromosome arms, and most of them were not DNA-methylated genes (S2F Fig).

To further dissect the genetic interaction between the DRM2, CMT3 and MAIN pathways, we generated the *drm1 drm2 main-3 (dd main-3)* and *cmt3 main-3* mutants (S2G Fig). We then analyzed the expression level of several TEs previously identified as misregulated in *ddc*, *main-3* and/or *ddc main-3*. The endogenous *ATCOPIA28* was the most expressed in *ddc main-3* and *dd main-3*, and to a lesser extent, in *cmt3 main-3* (Fig 2F). This is consistent with the fact that all the F2 *ddc #16* x *main-2* plants overexpressing *ATCOPIA28::GFP* were *drm2* homozygote, although they segregated the *cmt3* mutation (S1F Fig). Further analyses showed that most of the tested TEs tend to be more expressed in *cmt3 main-3* than in *dd main-3*, with the exception of *ATIS112A* that was more upregulated in *dd main-3* than in *cmt3 main-3* (Fig 2G). In conclusion, these analyses showed complex genetic interactions between the DRM2, CMT3 and MAIN pathways, suggesting that MAIN and DNA methylation pathways act synergistically to repress TEs and DNA-methylated genes.

## MAIN and MAIL1 are required for the proper expression of a common set of genes and TEs

Beside a role of MAIN in TE and gene silencing, our transcriptomic analyses using the hypomorphic *main-3* mutant suggested that MAIN would be required for the expression of several genes that are not controlled by the DRM2 and CMT3 pathway (S2E Fig). To further study the role of MAIN and MAIL1 in the regulation of gene expression and TE silencing, we performed

two independent RNA-seq experiments in the *main-2* and *mail1-1* null mutants (RNA-seq Exp1 and Exp3), and combined these experiments with the reanalysis of previously published RNA-seq datasets (RNA-seq Exp2) [15]. Principal component analyses (PCA) showed that for each RNA-seq experiment, *main-2* and *mail1-1* mutant samples tend to cluster together, and away from the WT samples (S3A Fig). Analyzing these three RNA-seq experiments together allowed to identify large numbers of genes and TEs that were misregulated in the *main-2* and *mail1-1* null mutants (Fig 3A and 3B and S2 Table).

We then compared the transcriptomes of *main-2* and *mail1-1* mutants, together with the *main-3* mutant allele (Fig 3A and 3B, S1 and S2 Tables). As expected by the fact that *main-2* and *mail1-1* are null mutants while *main-3* is a hypomorphic mutant allele, we identified greater numbers of misregulated loci in *main-2* and *mail1-1* in comparison to *main-3* (Fig 3A and 3B). Fractions of these loci were specifically misregulated in each mutant background (Fig 3C and 3D). In addition, we identified subsets of genes and TEs that were only misregulated in *main-2* and *mail1-1* null mutants, but not in the hypomorphic *main-3* mutant (Fig 3C and 3D and S3 Table). Finally, these analyses revealed subsets of loci that were commonly misregulated in the three mutant backgrounds (Fig 3C and 3D, S3B–S3D Fig and S3 Table).

The biggest overlaps between misregulated loci in *main-2*, *mail1-1* and *main-3* mutants were among the downregulated genes and upregulated TEs, whereas only a small proportion of genes commonly upregulated in *main-2* and *mail1-1* were also upregulated in *main-3* (Fig 3D). As observed in *main-3* (S2F Fig), upregulated TEs in *main-2* and *mail1-1* were enriched in pericentromeric regions, and genes that were downregulated in *main-2* and *mail1-1* were not targeted by DNA methylation, and mostly located in the chromosome arms (Fig 3E). However, unlike in *main-3*, the upregulated genes in *main-2* and *mail1-1* were not enriched in pericentromeric regions, and only small fractions of them were DNA-methylated genes (Fig 3E). This discrepancy can be explained by the fact that *main-2* and *mail1-1* null mutations have a much greater impact on the misregulation of gene expression than the hypomorphic *main-3* mutant allele.

Finally, we compared the sets of misregulated loci in *main-2*, *mail1-1*, *ddc* and *ddc main-3* (S1 and S2 Tables). We found significant overlaps among upregulated genes and TEs between *main-2*, *mail1-1*, *ddc* and *ddc main-3* (S3E Fig). This suggests that MAIN, MAIL1, DRM2 and CMT3 cooperate to silence these subsets of genes and TEs. However, we could not find significant overlaps among downregulated genes between *main-2*, *mail1-1* and *ddc* (S3E Fig). Instead, a significant overlap was identified only by comparing the lists of downregulated genes in *main-2*, *mail1-1* and *ddc main-3*, three genetic backgrounds carrying a mutation in either *MAIN* or *MAIL1* (S3E Fig). Thus, this suggests that MAIN and MAIL1 are required for the expression of specific genes, in a DRM2- and CMT3-independent manner.

In conclusion, these comparative analyses allowed to precisely define the loci that were misregulated in *main-2* and *mail1-1* in comparison to *main-3*, *ddc* and *ddcmain-3* mutants. Among these loci, several TEs and DNA-methylated genes are commonly targeted by the MAIN, MAIL1, DRM2 and CMT3 pathways, which suggests that MAIN, MAIL1 and DNA methylation pathways cooperate to silence these TEs and DNA-methylated genes. Besides, several genes are downregulated in *main-2* and *mail1-1*, and subsets of these genes are also downregulated in *main-3*, and *ddcmain-3* but not in *ddc*. This suggests that the MAIN and MAIL1 act independently of DRM2 and CMT3 to ensure the expression of these genes. Finally, these results revealed important overlaps between the misregulated loci in *main-2* and *mail1-1* null mutants, which strongly suggests that the two proteins act in the same pathway to regulate the expression of common sets of loci.

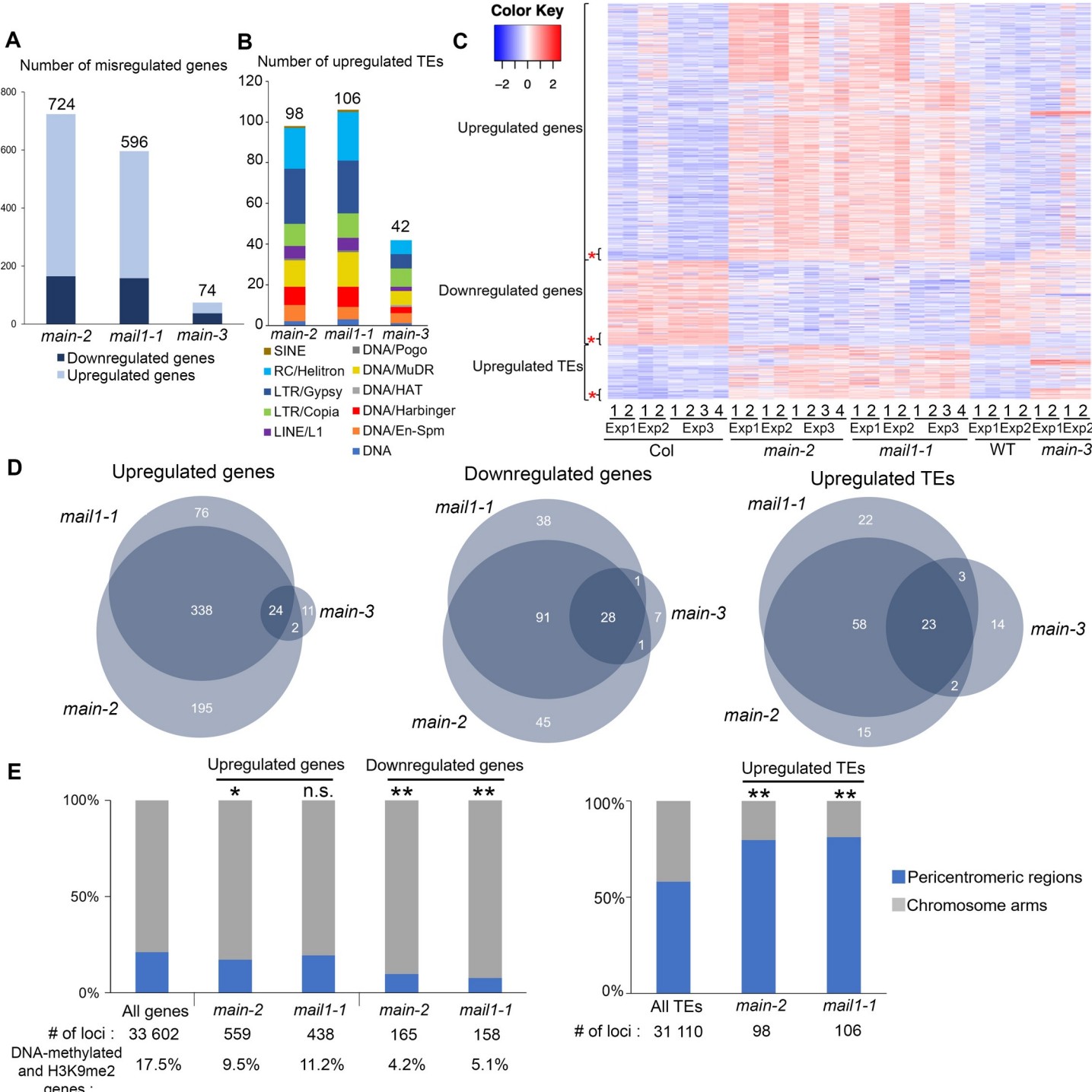

**Fig 3. MAIN and MAIL1 are required for the proper expression of similar genes, and for TE silencing.** (A-B) Number of misregulated genes (A) and upregulated TEs (B) in *main-2*, *mail1-1* and *main-3* mutants in comparison to WT Col plants. TEs are classified by superfamily. (C) Heatmap showing misregulated loci in *main-2*, *mail1-1* and *main-3* in comparison to Col and WT controls, respectively. Asterisks represents loci that are commonly misregulated in the three mutant backgrounds. (D) Venn diagrams analyses representing the overlaps between misregulated loci in *main-2*, *mail1-1* and *main-3*. Fisher's exact test statistically confirmed the significance of Venn diagram overlaps (p-value <2.2.10e-16). (E) Fraction of misregulated loci in *main-2* and *mail1-1* located in chromosome arms or in pericentromeric regions as defined in [50]. Asterisks indicate statistically significant enrichments of downregulated genes and upregulated genes and TEs in chromosome arms and pericentromeric regions, respectively, in comparison to the genomic distributions of all *A. thaliana* genes and TEs (Chi-Square test, *: p-value≤ 0.05, **: p-value≤ 0.01, n.s: not significant). Percentages of genes targeted by DNA methylation and H3K9me2 were calculated based on enrichment in heterochromatin states 8 and 9 as defined in [51]. RNA-seq threshold: log2≥2, or log2≤-2; p-adj< 0.01.

## Slight increase in non-CG methylation in the *main-2* mutant does not correlate with changes in gene expression and TE silencing defect

Whole genome bisulfite sequencing (BS-seq) analyses showed that, at the chromosome scale, DNA methylation level is mostly unchanged in *main-2* in comparison to WT, with the exception of a slight increase in CHG methylation in pericentromeric regions (Fig 4A). Subtle but statistically significant CHG hypermethylation was further confirmed in pericentromeric TEs and genes, which are mostly TE genes (Fig 4B and 4C). Slight CHG and CHH hypermethylation was also detected in TEs located in chromosome arms (Fig 4D). Conversely, genes located

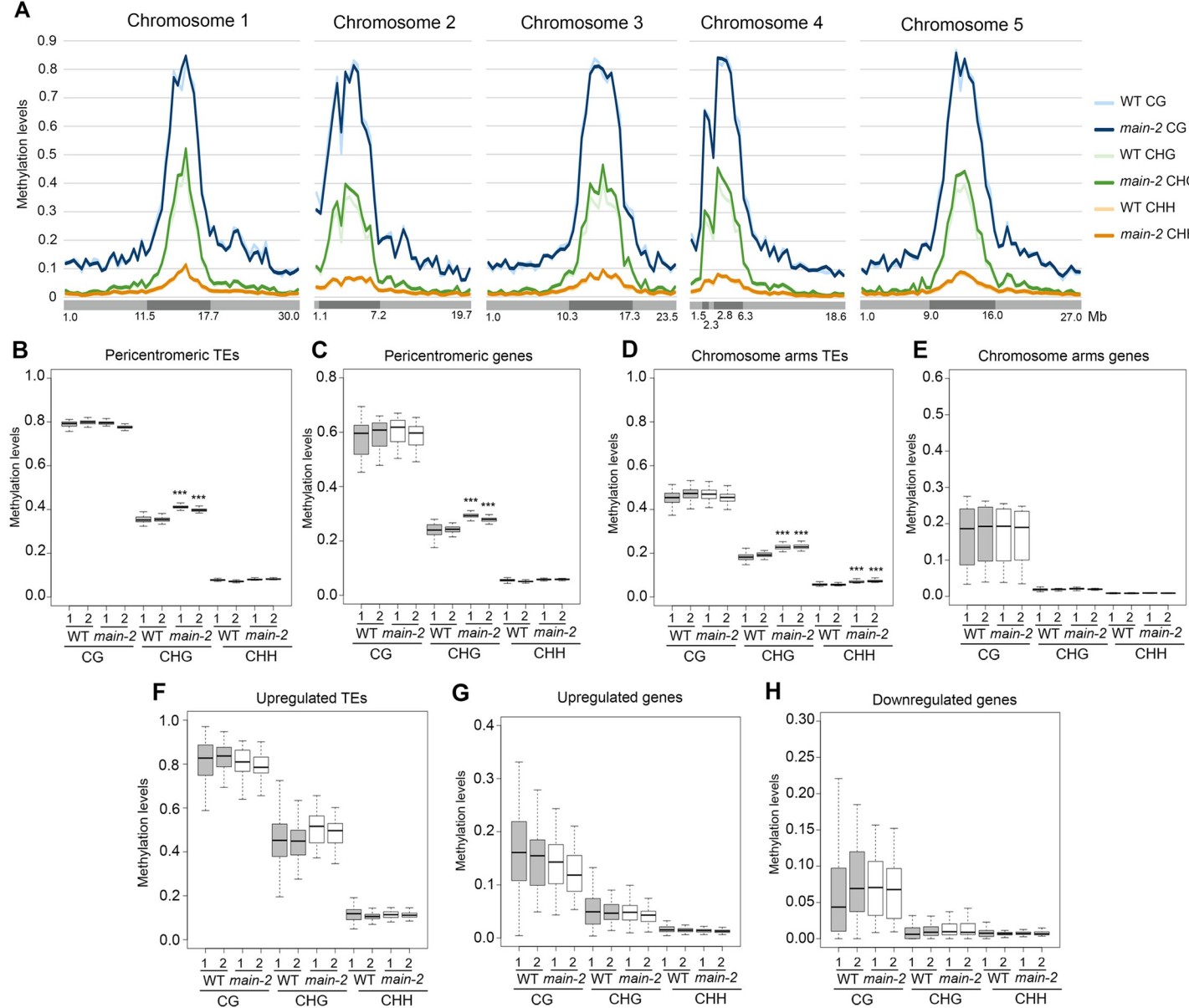

**Fig 4. The *main-2* mutation has a slight effect on non-CG DNA methylation levels.** (A) Genome-wide DNA methylation levels along the five *Arabidopsis* chromosomes in *main-2* versus WT Col plants. Chromosome arms are depicted in light grey, pericentromeric regions in dark grey as defined in [50]. Mb: megabase. (B-H) Boxplot analyses in two *main-2* and WT Col biological replicates showing the DNA methylation levels of all pericentromeric TEs (B) and genes (C), all chromosome arms TEs (D) and genes (E), TEs that are upregulated in *main-2* (F), and genes that are upregulated (G) and downregulated (H) in *main-2*. p-values were calculated using a Wilcoxon test. ***: p-value < 2.10e-16.

in chromosome arms did not show significant changes in DNA methylation level in *main-2* (Fig 4E). Identical results were obtained by analyzing the DNA methylation level at upregulated TEs and misregulated genes in *main-2* (Fig 4F–4H). We then analyzed the DNA methylation level at genomic locations previously defined as differentially hypomethylated regions (hypo DMRs) at CHG and CHH sites in *cmt3* and *drm1drm2 (dd)* mutants, respectively [26]. The *cmt3* and *dd* hypo DMRs are mostly located in TEs. As observed with pericentromeric genes and all TEs (Fig 4B–4D), we found slight increases in CHG and CHH methylation at *cmt3* and *dd* hypo DMRs, respectively, in *main-2* (S4A and S4B Fig). Finally, DMR calling in *main-2* using stringent parameters only identified a few DMRs (S4C Fig). Thus, DNA methylation is mostly unaffected in *main-2*, with the exception of a slight increase in non-CG methylation at pericentromeric genes and all TEs. Moreover, this subtle non-CG hypermethylation does not correlated with changes in gene and TE expression observed in *main-2* because DNA methylation level in *main-2* is unchanged at these misregulated loci (Fig 4F–4H).

## MAIN, MAIL1 and the metallo-phosphatase PP7L physically interact together

The *main-2* and *mail1-1* null mutants display similar molecular and developmental phenotypes (Fig 3 and Fig 5A). Thus, we hypothesized that MAIN and MAIL1 proteins may act in the same pathway, possibly by interacting together. To test this hypothesis, we generated transgenic lines expressing FLAG- and MYC-tagged genomic PMD versions driven by their endogenous promoters. We confirmed that epitope-tagged MAIN and MAIL1 proteins were produced at the expected sizes, and they could complement the respective developmental phenotypes of null mutant plants (Fig 5A and 5B). Importantly, they could also efficiently rescue the TE silencing and gene expression defects observed in *main-2* and *mail1-1* mutants, implying that epitope-tagged MAIN and MAIL1 are functional proteins (Fig 5C–5E). Using FLAG-tagged MAIN and MAIL1 expressing plants, immunoprecipitation followed by mass spectrometry (IP-MS) analyses were carried out to determine potential protein interactors. Mass spectrometry (MS) analyses indicated that MAIL1 was strongly immunoprecipitated with MAIN-FLAG and *vice versa* (Fig 5F). To validate IP-MS results, we crossed the MAIN-FLAG and MAIL1-MYC lines together. We then performed co-immunoprecipitation (co-IP) experiments using F1 hybrid plants co-expressing the two transgenes, and confirmed that MAIN and MAIL1 interact together (Fig 5G). MS analyses of MAIN-FLAG and MAIL1-FLAG IP also identified the metallo-phosphatase PP7L as putative interactor (Fig 5F). MAIN, MAIL1 and PP7L were the only three proteins reproducibly enriched across multiple replicates (Fig 5F). Co-IP experiments using plants co-expressing either PP7L-FLAG together with MAIN-MYC or MAIL1-MYC constructs confirmed the interaction between PP7L and each PMD protein (Fig 5H and 5I). Thus, the three proteins MAIN, MAIL1 and PP7L physically interact together.

## The *main*, *mail1* and *pp7l* mutants display similar developmental and molecular phenotypes

PP7L is a putative metallo-phosphatase that was recently identified as a nuclear protein required for photosynthesis [20, 25]. The *pp7l-2* null mutant displays abnormal developmental phenotype reminiscent of *main-2* and *mail1-1* mutant plants, and 3-week-old *mail1-1 pp7l-2* double mutant plants do not show exacerbation of this phenotype (Fig 6A). To determine the genetic interaction between PMD and PP7L, we compared the transcriptomes of *main-2*, *mail1-1*, *pp7l-2* single and *mail1-1 pp7l-2* double mutants (S5A Fig and S2 and S4 Tables). We identified large numbers of misregulated loci in *pp7l-2* and *mail1-1 pp7l-2* (S5B and S5C Fig).

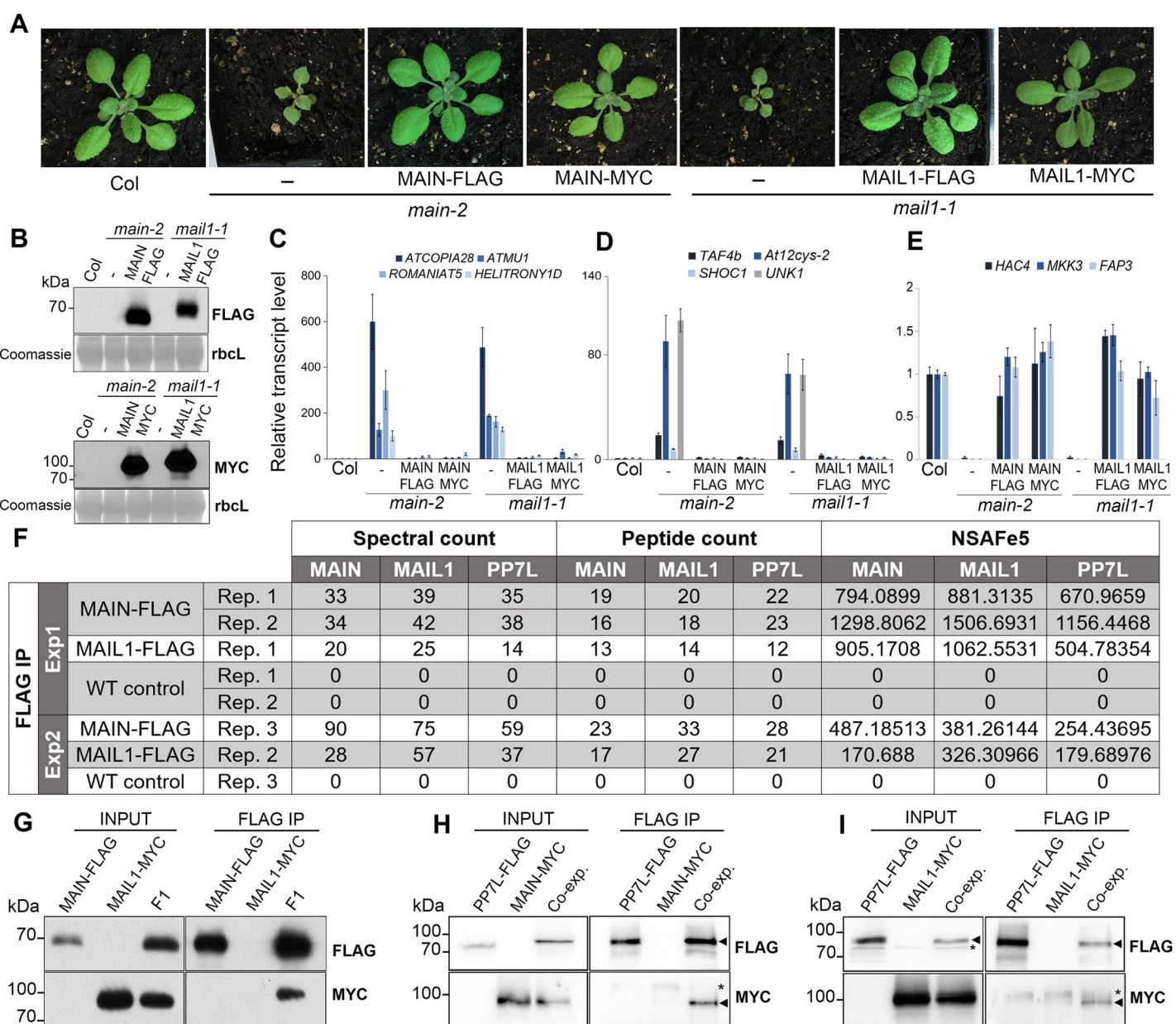

**Fig 5. MAIN, MAIL1 and PP7L physically interact together.** (A) Representative pictures of 3-week-old *main-2* and *mail1-1* mutants, and epitope-tagged complementing lines in comparison to WT Col plants. (B) Western blots using anti-FLAG and anti-MYC antibodies showing the accumulation of epitope-tagged PMD proteins at the expected sizes in the different complementing lines. Coomassie staining of the large Rubisco subunit (rbcL) is used as a loading control. KDa: kilodalton. (C-E) Relative expression analyses of upregulated TEs (C), upregulated genes (D) and downregulated genes (E) in the different complementing lines assayed by RT-qPCR. RT-qPCR analyses were normalized using the housekeeping *RHIP1* gene, and transcript levels in the complementing lines and mutants are represented relative to WT Col. Error bars indicate standard deviation based on three independent biological replicates. (F) FLAG-tagged MAIN and MAIL1 proteins were immunoprecipitated and putative interacting proteins were identified by mass spectrometry. Numbers of identified spectra, peptides and the normalized spectral abundance factor (NSAFe5) are shown for two independent experiments, including three *main-2* and two *mail1-1* replicates. WT replicates are used as a negative control. Only proteins reproducibly enriched in all the FLAG-MAIN and FLAG-MAIL1 IP, and depleted in WT controls across multiple replicates are described in the table. (G) MAIL1-MYC was co-immunoprecipitated with MAIN-FLAG in F1 plants obtained by crossing MAIL1-MYC and MAIN-FLAG lines together. Parental MAIL1-MYC and MAIN-FLAG lines were used as negative controls. (H) The MAIN-MYC line was supertransformed with the PP7L-FLAG construct, and MAIN-MYC was co-immunoprecipitated with PP7L-FLAG. Plants expressing only MAIN-MYC or PP7L-FLAG were used as negative controls. (I) Same as H but using MAIL1-MYC plants supertransformed with the PP7L-FLAG construct. Epitope-tagged proteins were detected by Western blotting. Arrowheads indicates expected bands. Asterisks indicates non-specific hybridization. Co-exp: plants co-expressing PP7L-FLAG and MAIN-MYC (H) or PP7L-FLAG and MAIL1-MYC (I).

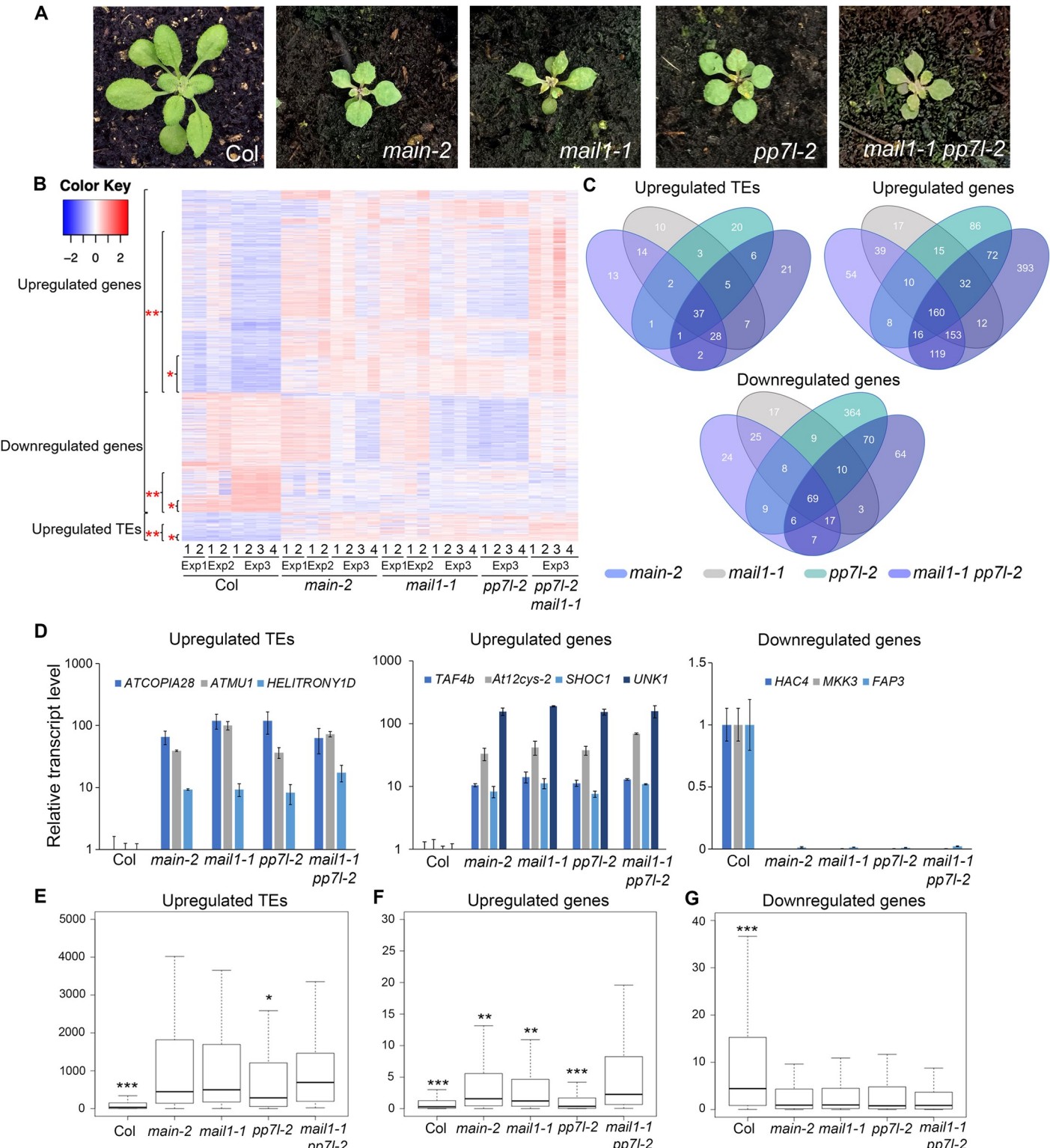

**Fig 6. *main-2, mail1-1, pp7l-2* single and *mail1-1 pp7l-2* double mutants display similar developmental and molecular phenotypes.** (A) Representative pictures of 3-week-old *main-2*, *mail1-1*, *pp7l-2* single and *mail1-1 pp7l-2* double mutants in comparison to WT Col plant. (B) Heatmap showing misregulated loci in *main-2*, *mail1-1*, *pp7l-2* and *mail1-1 pp7l-2* mutants in comparison to WT Col plants using the datasets of RNA-seq Exp1, Exp2 and Exp3 (S2 and S4 Tables). One asterisk defines the loci that are commonly misregulated in all mutant backgrounds. Two asterisks define the loci that are misregulated in the *mail1-1 pp7l-2* double mutant. (C) Venn diagrams analyses representing the overlaps between misregulated loci in *main-2*, *mail1-1*, *pp7l-2* and *mail1-1 pp7l-2*. Fisher's exact test statistically

confirmed the significance of Venn diagram overlaps (p-value <2.2.10e-16). (D) Relative expression analyses of upregulated TEs, genes and downregulated genes in the different genotypes assayed by RT-qPCR. RT-qPCR analyses were normalized using the housekeeping *RHIP1* gene, and transcript levels in the different mutants are represented relative to WT Col. Error bars indicate standard deviation based on three independent biological replicates. (E-G) Boxplots analyses showing average RPKM values of upregulated TEs (E), upregulated genes (F) and downregulated genes (G) in *mail1-1 pp7l-2* in the indicated genotypes of RNA-seq Exp3. These analyses are based on the misregulated loci datasets defined by ** in panel B. P-values were calculated using a Wilcoxon test, and only significant p-values are shown. *: p-value< 1.10e-3; **: p-value < 3.10–6; ***: p-value< 2.10e-16.

As observed in *main-2* and *mail1-1*, TEs upregulated in *pp7l-2* and *mail1-1 pp7l-2* were enriched in pericentromeric regions, while up- and downregulated genes were mostly located in the chromosome arms, and not targeted by DNA-methylation (S5D Fig).

Comparative analyses revealed that significant proportions of loci were commonly misregulated in *main-2*, *mail1-1*, *pp7l-2* and *mail1-1 pp7l-2* mutants, which is consistent with the fact that MAIN, MAIL1 and PP7L interact together to possibly regulate gene expression and silence TEs (Fig 6B–6D and S5 Table). These analyses also identified loci that were specifically misregulated in *main-2*, *mail1-1* or *pp7l-2*, which suggests that each protein is independently required for the proper expression of subsets of loci (Fig 6B and 6C). Besides, these analyses revealed loci that were exclusively misregulated in the *mail1-1 pp7l-2* double mutant, which implies that PP7L and MAIL1 may act redundantly to ensure the proper expression of these loci (Fig 6B and 6C). Further analyses showed that, among the loci that were misregulated in *mail1-1 pp7l-2*, upregulated genes were significantly more expressed in the double mutant than in each single mutant, and upregulated TEs were significantly differentially expressed only between *mail1-1 pp7l-2* and *pp7l-2* mutants (Fig 6E and 6F). Conversely, there was no significant difference of expression between the double mutant and single mutants for the downregulated genes (Fig 6G). Thus, these analyses suggest that combining the *pp7l-2* and *mail1-1* mutations may lead to synergistic defects mostly at genes that are upregulated in the double mutant.

We then performed in silico analyses to identify enriched DNA motif within a 1kb promoter region upstream of start codon of genes that were up- or downregulated in the different mutant backgrounds. We could not detect any enrichment of a DNA motif among any lists of upregulated genes (including overlapping lists). Likewise, we could not identify a DNA motif enriched in the lists of downregulated genes in *pp7l-2* or *ddc*. However, we identified a discrete DNA motif (hereafter called 'DOWN' motif) that was partially enriched in the promoter of genes that were downregulated in *main-2*, *mail1-1* and *mail1-1 pp7l-2* mutants (S5E Fig). The *main-2*, *mail1-1*, *pp7l-2* and *mail1-1 pp7l-2* null mutants display strong developmental phenotype, and large numbers of misregulated loci (Fig 6A). Therefore, it is likely that some of the gene misregulation observed in these mutants might be due to side effects of the mutations. To overcome this issue and refine our analysis, we investigated the proportion of the 'DOWN' motif among downregulated genes in the hypomorphic *main-3* and *ddc main-3* mutants, as well as in the different overlapping lists of genes commonly downregulated (S3 and S5 Tables). The 'DOWN' motif was strongly enriched among the downregulated genes in *main-3*, and to a lesser extent in *ddc main-3* (S5E Fig). It was also significantly enriched in the overlapping lists of commonly downregulated genes in *main-2*, *mail1-1* and *main-3* as well as in the *main-2*, *mail1-1*, *pp7l-2* and *mail1-1 pp7l-2* overlap (S5E Fig). It was further enriched in the promoters of genes commonly downregulated in all the mutant backgrounds—except *ddc*—analyzed in this study: twenty-five out of twenty-six genes, 96% of enrichment (S5E Fig, S6 and S7 Tables). We analyzed the DNA methylation level of the 'DOWN' motif in the promoters of these twenty-five genes in WT and *main-2*, and found that this DNA motif was not targeted by DNA methylation. Besides, further analyses showed that only a small fraction of all *Arabidopsis* genes carried the 'DOWN' motif in their promoter (12,46%, S5E Fig). Finally, random test

analyses based on twenty-six randomly picked genes strongly suggested that the enrichment of the 'DOWN' motif in the promoter of downregulated genes was substantial (S7 Table).

Thus, altogether, these analyses showed that MAIN, MAIL1 and PP7L are equally required for the repression of several genes and TEs. The three proteins are also required for the proper expression of a common set of genes that are downregulated in each single mutant as well as in *mail1-1 pp7l-2* double mutant, and significant fractions of these downregulated genes carry the 'DOWN' DNA motif in their promoter. Furthermore, the 'DOWN' DNA motif is strongly enriched among the genes that are always identified as downregulated in every mutant background carrying the *main-2*, *mail1-1*, *pp7l-2* or *main-3* mutant alleles. This suggests that transcriptional activation of this subset of loci equally requires MAIN, MAIL1 and PP7L activity, and possibly the recognition of the 'DOWN' DNA motif.

## PP7L is not required for heterochromatin condensation

WT *Arabidopsis* nuclei at interphase exhibit condensed DNA foci called chromocenters that are composed of constitutive heterochromatin, and are enriched in H3K9me2 [27]. In several epigenetic mutants, decondensation of constitutive heterochromatin correlates with disruption of chromocenters, and loss or diffusion of H3K9me2 in the nucleoplasm [27]. Thus, analyzing H3K9me2 subnuclear distribution by immunofluorescence (IF) experiments has been reproducibly used as a cytological approach to assay for heterochromatin decondensation [12, 27, 28]. A previous study showed that subnuclear distributions of chromocenters and H3K9me2 were unchanged in *main-2* and *mail1-1* mutants [15]. However, fluorescent in situ hybridization (FISH) experiments using a DNA probe for the 106B pericentromeric repeats suggested that heterochromatin was decondensed in the *main-2* and *mail1-1* in comparison to WT plants [15]. We performed IF experiments to analyze the subnuclear distribution of H3K9me2 in the *pp7l-2* mutant. These analyses did not show any change in the condensation level of chromocenters in *pp7l-2* nuclei in comparison to WT (Fig 7). Instead, we observed that *pp7l-2* nuclei were proportionally more condensed than WT nuclei (Fig 7). This is likely due to the fact that *pp7l-2* mutant displays abnormal phenotype and growth delay in comparison to WT plants that are entering the floral transition stage, a developmental stage where partial chromocenter decondensation has been documented [29]. In conclusion, based on the H3K9me2 IF experiments, we can conclude that *pp7l-2* is not impaired in chromocenter condensation.

## The PMD and PP7 domains have co-evolved among the Eudicots

Among the Angiosperms, most of the genic PMDs, like MAIN and MAIL1, are standalone versions [18]. However, some genic PMDs can associate with other protein domains, such as for instance a PPP domain. In *A. thaliana*, the protein MAIL3, which carries a PMD fused to a PPP domain, is a close homolog of both MAIN/MAIL1 and PP7/PP7L through its PMD and PPP domains, respectively. Considering that the PMD proteins MAIN and MAIL1 interact with PP7L, and are required for the expression of similar set of loci, we decided to determine the distribution of related genic PMD and PPP domains, and to retrace their evolutionary history among plant species. The *A. thaliana* MAIN, MAIL1 and MAIL3 are all members of the PMD-C family that also includes MAIL2 [15]. Since our objective is to retrace the evolution of genic (and not TE-containing) PMD-C, we have decided to restraint our search to Eudicots. Indeed, Eudicot species contain mainly genic PMD-C, while other angiosperms may contain variable numbers of closely related genic and TE-associated PMD-C motifs that would be difficult to distinguish in our analysis. To retrace the evolution history of the genic PMD-C family, we used *A. thaliana* PMD-C genes to search and collect their relatives (paralogues and

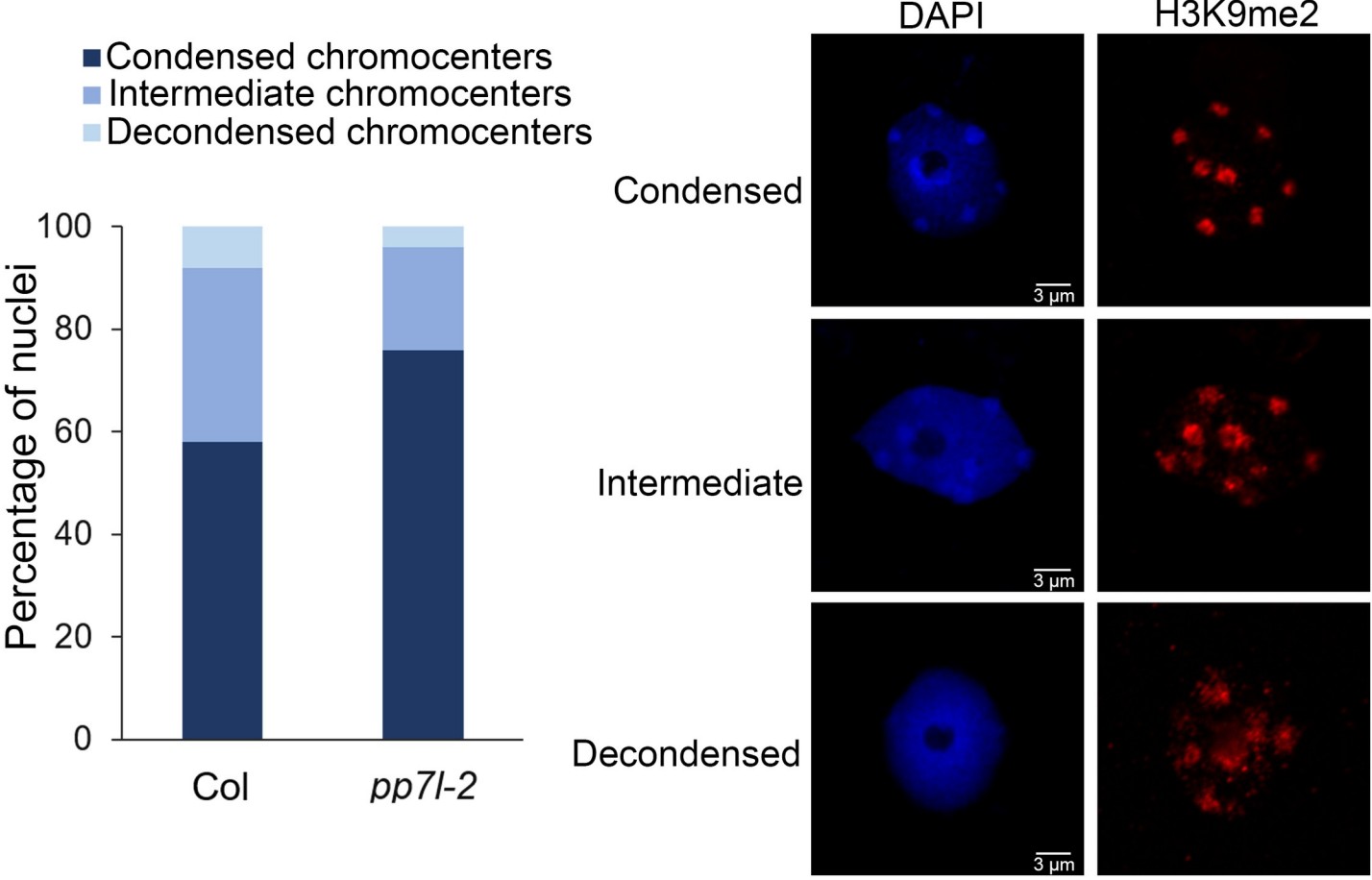

**Fig 7. Constitutive heterochromatin appears unaltered in *pp7l-2* mutant.** Proportion of nuclei showing condensed, partially decondensed (intermediate), or decondensed chromocenters in the *pp7l-2* mutant in comparison to WT control (Col) based on H3K9me2 immunostaining of nuclei. Representative pictures of nuclei displaying condensed, partially decondensed or decondensed chromocenters. DAPI: DNA stained with 4′,6-diamidino-2-phenylindole.

orthologues) in 30 genomes representative of the Eudicot diversity (see S8 Table for a list of species and their corresponding codes used in Fig 8 and S9 Table for motif sequences).

In our phylogenetic analysis, the genic PMD-C family can be clearly separated in two major clades. The first clade is composed of orthologues of *A. thaliana* MAIL2, MAIL1 and MAIN, while the second one includes orthologues of *A. thaliana* MAIL3 (Fig 8A). MAIL2 orthologues were found in all species tested, forming a closely related group, which suggests that they are under strong purifying selection (see the very short branch lengths linking most MAIL2 genes in Fig 8A). In several species, additional MAIL2 paralogues were also detected. They were either imbedded in the major MAIL2 group, or forming independent and more divergent subgroups, like in the case of MAIL1 and MAIN that are Brassicaceae-specific MAIL2 paralogues. By comparison, MAIL3 orthologues were not found in all Eudicot species tested, and, except in Brassicaceae, *MAIL3* genes appear to be under much weaker purifying selection compare to *MAIL2* and *MAIL2-like* genes (see the longer branch lengths in the tree of Fig 8A). Brassicaceae *MAIL3* genes contrast with other *MAIL3*, by forming a closely related group in the phylogenetic tree. This suggests a clear change in selection pressure, typical of a neofunctionalization event that could correlate with the acquisition of the PPP motif by these genes (Fig 8B and see below). Remarkably, another fusion event between PMD-C and PPP motifs

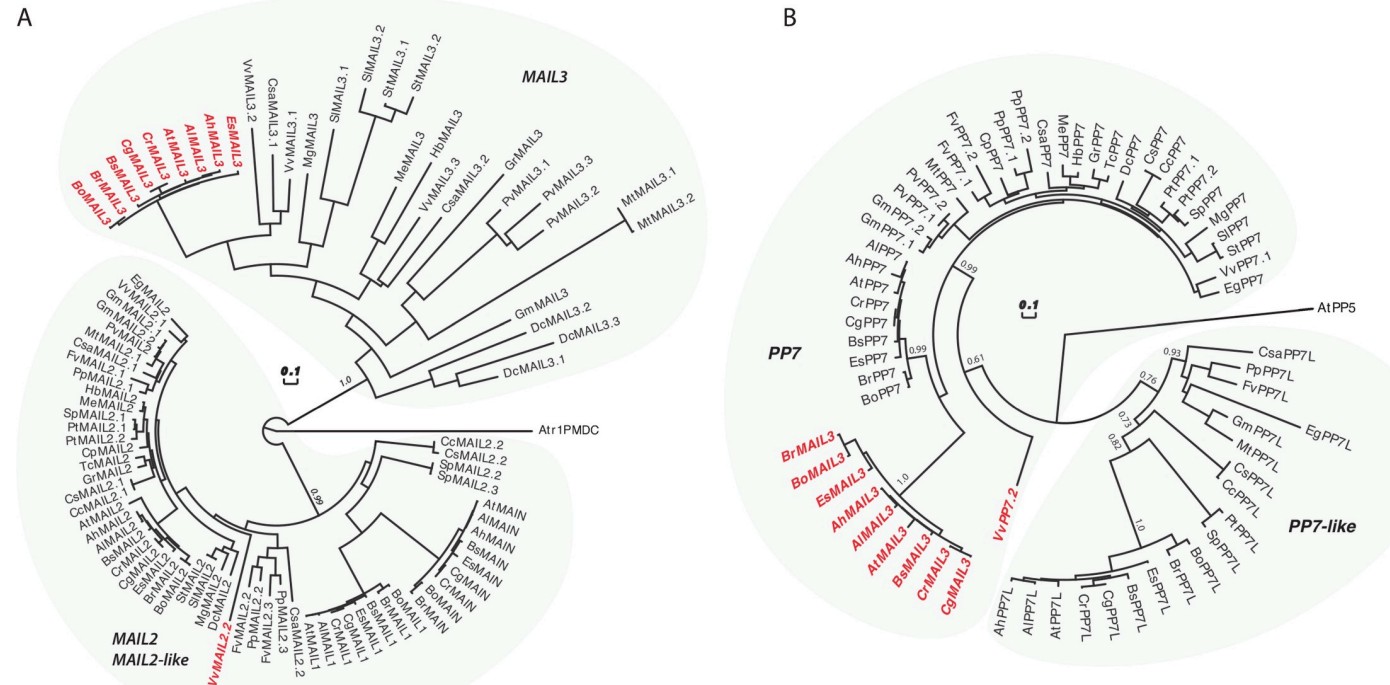

**Fig 8. Evolutionary history of PMD-C and PP7 proteins in plants.** (A) An alignment of the PMD-C motifs from 30 representative Eudicot species was used to construct a phylogenetic tree. The two major clades (MAIL2/MAIL2-like and MAIL3) are indicated. The species codes are given in S11 Table, and corresponding protein sequences in S12 Table). In red are genes presenting a fusion between a PMD-C and a PP7 motif. Statistical supports of key nodes calculated with the approximate likelihood-ratio test are indicated. Scale bar indicates one substitution/site. The tree was rooted using the *Amborella trichopoda* PMD-C motif (Atr1PMDC). (B) Phylogenetic tree constructed using an alignment of the PP7 motif from the same species as in (A). The two major clades (PP7 and PP7L) are indicated. In red are genes presenting a fusion between a PP7 and a PMD-C motif. Statistical supports of key nodes calculated with the approximate likelihood-ratio test are indicated. Scale bar indicates one substitution/site. The tree was rooted using the *A. thaliana* PP5 motif (AtPP5).

occurred independently in grapevine, but this time involving a MAIL2 paralogue (VvMAIL2.2, Fig 8A).

We then used the PPP motif found in *A. thaliana* MAIL3, to collect orthologous genes and retrace the evolution history of this motif in the same Eudicot species used above. We confirmed that these genes can be clearly separated in two distinct clades: PP7 and PP7-like (PP7L) (Fig 8B). All tested species present one or several closely related PP7 paralogues. Although the Brassicaceae MAIL3 PPP motif belongs to the PP7 clade, it diverged significantly compared to other standalone PP7 paralogues (Fig 8B). Same observation was made regarding the PP7 domain of VvMAIL2.2. Thus, as described for the PMD of Brassicaceae MAIL3 and grapevine VvMAIL2.2, this suggests a fast-evolving period and neofunctionalization of the PP7 domain in these species, subsequently to the PMD-C/PP7 fusion. Conversely, PP7L orthologues were not found in all species tested and, accordingly, these genes are under weaker purifying selection compare to genes belonging to the PP7 subfamily. In conclusion, phylogenetic analyses showed that, in at least Brassicaceae and grapevine, neo-association of PMD-C and PP7 domains have potentially create new protein functions that were maintained through evolution.

## Discussion

In *A. thaliana*, MAIN and MAIL1 are standalone PMD proteins that have been involved in genome integrity, regulation of cell division and differentiation, and silencing of TEs [15–17].

In this study, we show that TE silencing is widely impaired in the *ddc main-3* higher order mutant, which is both partially defective in DNA methylation and MAIN activity. We also identify the putative phosphatase protein PP7L as MAIN and MAIL1 protein interactor, and show that among the loci that are commonly misregulated in *pmd* and *pp7l* single and double mutants, a substantial fraction of downregulated genes carries the 'DOWN' DNA motif in their promoter. Finally, phylogenetic analyses among Eudicots suggest a mechanism of neo-functionalization between the PMD and PP7-type PPP, to potentially acquire a functional module that requires the two protein domains.

## The PMD MAIN protein acts independently of DRM2- and CMT3 pathways to silence TEs and DNA-methylated genes

Previous analyses showed that some TEs were synergistically upregulated in the *mail1 rdr2* double mutant plants, suggesting that MAIL1 acts independently of RdDM pathway [15]. In our whole genome transcriptomic analyses, we show that several TEs and DNA-methylated genes are upregulated in both *main-3* and *ddc* mutants, as well as in the *ddc main-3* quadruple mutant (Fig 2 and S2 Fig). We also identify TEs that are upregulated in either *ddc* or *main-3* mutants, but display stronger misregulation in the *ddc main-3* higher order mutant (Fig 2 and S2 Fig). Finally, we identify a large class of TEs that are only upregulated in *ddc main-3* (Fig 2 and S2 Fig). Altogether, these analyses reveal complex genetic interaction between the MAIN, DRM2 and CMT3 proteins to silence TE. Previous work showed that DNA methylation is not impaired in *mail1-1* [15]. We found that DNA methylation is mostly unaffected in the *main-2* null mutant. However, we detected a mild but significant hypermethylation at non-CG sites in TEs and pericentromeric genes (Fig 4). One hypothesis is that CHG and CHH hypermethylation observed in *main-2* is a backup mechanism to compensate for MAIN loss of function, and to dampen TE silencing defects. Although further studies will be required to test this hypothesis, it is consistent with the fact that combining the *main-3* and *ddc* mutations leads to an exacerbation of TE silencing defects. Thus MAIN, DRM2 and CMT3 pathways cooperate to silence TE. Synergistic effects between different epigenetic pathways have already been described. For instance, it has been shown that MORPHEUS MOLECULE 1 (MOM1) and MORC1/MORC6 proteins, or MOM1 and the RdDM pathway act synergistically to efficiently silence TEs [13, 30]. Altogether, these observations contribute to the "mille-feuille" (i.e. "multiple layers") model, in which different epigenetic pathways converge towards the silencing of TEs [31].

## The putative phosphatase PP7L interacts with the PMD MAIN and MAIL1 protein to regulate a similar set of genes and TEs

Recently, the putative phosphoprotein phosphatase PP7L was involved in the biogenesis of chloroplasts and plant response upon abiotic stress [25]. Here, we show that PP7L interact with MAIN and MAIL1, and *main-2*, *pp7l-2*, *mail1-1* single and *mail1-1 pp7l-2* double mutant plants display similar developmental and molecular phenotypes (Figs 5 and 6). We also show that, as described for *main-2* and *mail1-1* [15], the subnuclear distribution of chromocenters and H3K9me2 are unaltered in *pp7l-2* (Fig 7). The 106B pericentromeric repeats appeared decondensed in *main-2* and *mail1-1* mutants [15], future work will determine if similar phenotype is observed in *pp7l-2*. Although MAIN, MAIL1 and PP7L interact together, we cannot exclude that an additional protein is required for the interaction. In addition, PP7L may have additional partners independently of MAIN and MAIL1. Further biochemical studies such as IP-MS analyses using the FLAG-tagged PP7L line will contribute to addressing these points.

Transcriptomic analyses revealed complex genetic interaction between MAIN, MAIL1 and PP7L; the three proteins acting either independently or together to ensure the proper expression of genes, and to perform TE silencing. Moreover, transcriptome profiling of *mail1-1 pp7l-2* double mutant revealed that the two mutations may have synergistic effects, specifically at genes that are upregulated in the mutant. To further study the genetic interaction between the three proteins, it will be important to analyze the transcriptome of *main-2 mail1-1 pp7l-2* triple mutant. Altogether and considering that *i)* MAIN, DRM2 and CMT3 pathways cooperate to silence TEs, and *ii)* the *main-2* mutant show a slight increase in DNA methylation at CHG and CHH sites, we cannot rule out that MAIN is playing a dual role: regulating gene expression through its interaction with MAIL1 and PP7L, and involved in TE silencing through its genetic interaction with DNA methylation. In the future, it will be important to analyze DNA methylation in *pp7l-2*, but also in *pmd pp7l-2* higher order mutants. In parallel, studying the *ddc pp7l-2* mutant will allow to further decipher the genetic interaction between the PP7L and DNA methylation pathways.

### A fraction of genes that are commonly downregulated in *main*, *mail1* and *pp7l* mutants carry the 'DOWN' motif in their promoters

A substantial fraction of genes that are commonly downregulated in *main-2*, *mail1-1*, *pp7l-2* and *mail1-1 pp7l-2* carry the 'DOWN' motif in their promoter (S5E Fig and S7 Table). Furthermore, twenty-five out of twenty-six genes commonly downregulated in all the mutant backgrounds analyzed in this study—except *ddc*—carry the 'DOWN' DNA motif in their promoter (S5E Fig and S7 Table). The 'DOWN' motif is also enriched in fractions of downregulated genes in *main-2*, *mail1-1*, *mail1-1 pp7l-2*, *main-3* and *ddc main-3*. However, it is not enriched among downregulated genes in *pp7l-2* mutant. One explanation for this discrepancy is that too many loci were identified as downregulated in *pp7l-2*, which created a dilution of the loci carrying the 'DOWN' motif in their promoter.

Based on our results, we hypothesize that the 'DOWN' motif may act as a putative cis-regulatory element (CRE) recognized by an unidentified TF, which would be required for the transcription of genes identified as downregulated in *pmd* and *pp7l* mutants. This unknown TF could be recruited or activated by the MAIN/MAIL1/PP7L protein complex. Another hypothesis is that the 'DOWN' motif is directly recognized by the MAIN/MAIL1/PP7L protein complex. Further study will be required to test if MAIN/MAIL1/PP7L protein complex interact with chromatin, and bind the 'DOWN' motif. In parallel, further biochemical analyses may allow to identify an uncharacterized putative TF as MAIN/MAIL1/PP7L protein interactor.

Altogether, these analyses suggest that MAIN, MAIL1 and PP7L are involved in three distinct activities. First, they are required for the silencing of TEs and DNA-methylated genes, cooperating with canonical epigenetic factors such as DRM2 and CMT3 to efficiently repress these loci. Second, they are required for the repression of subsets of genes that are not targeted by DNA methylation. For this category of loci, one hypothesis is that MAIN, MAIL1 and PP7L may act as transcriptional repressor. Third, MAIN, MAIL1 and PP7L are required for the transcriptional activation of several genes, and fractions of those genes carry the 'DOWN' motif in their promoter. In the future, it will be important to determine the molecular mechanisms that are involved in these three activities of MAIN, MAIL1 and PP7L.

### The association of PMD-C and PP7/PP7L domains creates a functional protein module

In this study, we identified PP7L as a protein partner of the two standalone PMDs MAIN and MAIL1, and showed that these proteins are required for the proper expression of a common

set of genes, and for TE silencing. Besides, we showed that the Brassicaceae MAIL3 and the grapevine VvMAIL2.2 proteins carry a PMD fused to a PP7 domain. Based on these results, we hypothesize that depending on the configuration, the association of PMD-C and PP7/PP7L domains would create a functional protein module in trans or in cis. It is likely that the cis-association of PMD and PP7 found in the Brassicaceae MAIL3 proteins occurred in the common ancestors of this Eudicot lineage, possibly through the process of gene duplication. Since then, the MAIL3 PMD/PP7 fusion was maintained under strong purifying selection, arguing for a neofunctionalization of the fusion protein. It is likely that a similar process happened in grapevine, and possibly, in closely related Vitaceae species. To some extent, the two distinct events that occurred in Brassicaceae and grapevine are reminiscent of convergent evolution processes leading to the production of a functional PMD/PP7 module.

The occurrence of PMD and PP7/PP7L protein fusion in several Brassicaceae and grapevine is reminiscent of the concept of Rosetta stone chimera proteins, which describes that two proteins interacting together in one organism can be found fused together in another species to facilitate enzymatic activity [32]. There are several examples of Rosetta stone proteins, described for instance with different subunits of DNA topoisomerase or RNA polymerase [32]. Here, we show that, at least in *A. thaliana*, the Rosetta stone chimera MAIL3 coexist with its close homologs MAIN/MAIL1 and PP7L that interact together. The fact that the PMD and PP7 domains are fused together in MAIL3 may be a strategy to optimize protein activity. Conversely, the enzymatic activity of the MAIN/MAIL1/PP7L protein complex could be further regulated by allowing, or not, the three proteins to interact together. Nevertheless, in both scenarios, it is likely that PMD and PP7/PP7L association creates a functional protein module, which might be specialized in distinct biological processes depending on its composition. Thus, we hypothesize that the MAIL3 and MAIN/MAIL1/PP7L protein complexes play different role in the plant. This is consistent with the fact that, unlike *main-2*, *mail1-1* and *pp7l-2* mutant, the *mail3-2* mutant does not show abnormal developmental phenotype [17]. Further studies will be required to describe the role of MAIL3 in the plants.

In conclusion, we show here that the two *A. thaliana* PMD MAIN and MAIL1 proteins interact with PP7L, and are involved in the regulation of a common set of genes and TEs. In addition, we show that distinct events of PMD-C and PP7 fusions have occurred among the Eudicots (among several Brassicaceae species and in grapevine), suggesting some convergent evolution processes and a potential neofunctionalization of PMD/PP7 module in cis. The biological significance of PMD/PP7 fusion proteins will be investigated in the future by studying the role of MAIL3 in *A. thaliana*. In addition, it will be important to determine whether the PMD proteins play important roles in other plant species with agronomic value.

## Materials and methods

### Plant material and growing conditions

All the plant material is in the Columbia (Col) ecotype. Col = Non-transgenic WT Columbia ecotype. The *drm1-2* (SALK_031705), *drm2-2* (SALK_150863), *cmt3-11* (SALK_148381), *ddc* triple, *main-2* (GK-728H05), *mail1-1* (GK-840E05) and *pp7l-2* (SALK_003071) null mutant lines were previously described [15–17, 25, 26], and obtained from The Nottingham Arabidopsis Stock Centre. The *mail1-1 pp7l-2* double mutant was obtained by crossing the respective single mutants. T-DNA insertions were confirmed by PCR-based genotyping and RT-qPCR analyses. The *ATCOPIA28*::*GFP* WT line (WT) carries the transgene in WT Col ecotype. The *ATCOPIA28*::*GFP ddc* line (*ddc*) carries the transgene in *ddc*. The *ATCOPIA28*::*GFP ddc main-3* line (*ddc main-3 = ddc #16*) carries the transgene in the *ddc main-3* background. The *ATCOPIA28*::*GFP main-3* line (*main-3*) was obtained by backcrossing *ddc main-3* with WT,

F1 plants were self-fertilized, and F2 plants were screened by PCR-based genotyping to identify plants homozygote for the *main-3* mutation and WT for *DRM2* and *CMT3*. The *main-3* mutant allele was scored by derived cleaved amplified polymorphic sequences (dCAPS) using the restriction enzyme FokI. Primer sequences are described in S10 Table. All the WT Col and T-DNA mutant plants were grown on soil under a 16h-light/8h-dark cycle. When experiments required to screen for GFP expression under UV light, plants carrying the *ATCOPIA28*::*GFP* transgene were first grown on Murashige and Skoog (MS) plates under continuous light, 10-day old plants were then screened for GFP expression under UV light, and subsequently transferred onto soil. For *in vitro* plant culture, seeds were surface-sterilized and sowed on solid MS medium containing 0.5% sucrose (w/v).

## Cloning of ATCOPIA28::GFP

The pCambia3300-NLS-GFP-T35S vector was previously described [12]. The 5'LTR promoter corresponding to a region of ~1 kb upstream of *ATCOPIA28 (AT3TE51900)* was PCR ampli-fied from WT genomic DNA, and cloned into pCR2.1 TOPO vector (Invitrogen). Quikchange site-directed mutagenesis (Stratagene) was performed according to Manufacturer's instruction to create a polymorphism site (MfeI→NdeI) within the 5'LTR promoter, which was subse-quently mobilized into pCambia3300 upstream of NLS-GFP-T35S sequence. *ddc* triple mutant plants were transformed with the AT*COPIA28*::*GFP* construct using the *Agrobacterium*-medi-ated floral dip method [33]. Transgenic plants showing GFP fluorescence were backcrossed with a WT plant to promote the silencing of *ATCOPIA28*::*GFP* in the F1 generation. F1 plants were self-crossed and their F2 progenies were screened for GFP fluorescence, and PCR-based genotyped to obtain *ATCOPIA28*::*GFP* WT and *ATCOPIA28*::*GFP ddc* plants. Primer sequences used for *ATCOPIA28*::*GFP* cloning and PCR genotyping are described in S10 Table.

## EMS mutagenesis, GFP screening and mapping analyses

Five thousand seeds of *ATCOPIA28*::*GFP ddc* were mutagenized in 0.26% EMS solution for 12 hours with rotation. Seeds were subsequently washed with water and sown on soil. Fifteen hundred M2 populations were collected, and subsequently screened for GFP fluorescence under UV light using a SMZ18 Nikon Fluorescence Stereomicroscope coupled with the C-HGFI intensilight fluorescence filter. Pictures were taken using the DS Qi1MC digital cam-era kit. Mapping and identification of the EMS mutation responsible for the phenotype were performed by bulk segregant analysis coupled with deep genome re-sequencing as previously described [12], with the following differences. Reads were mapped against the reference genome (*Arabidopsis* TAIR10) and single nucleotide polymorphisms called in Geneious (Bio-matters). Using R, single nucleotide polymorphisms were filtered for EMS mutations (G: C→A:T) and zygosity called based on the variant frequency provided by Geneious ($\geq$80% homozygous mutation, $\geq$45%, and $\leq$55% heterozygous mutation). Plots were then created by calculating the ratio of the number of homozygous and heterozygous and mutations in a 500-kb window as previously described [34].

## Cloning of epitope-tagged versions of PMD and PP7L proteins

*MAIN*, *MAIL1* and *PP7L* genomic regions were PCR amplified and FLAG or MYC epitopes were added to the C-terminus of each protein as previously described [12]. Each time, the amplified region includes a ~1Kb promoter sequence upstream of the respective transcrip-tional start site. Within the *MAIN* promoter, a MluI site was modified to allow LR reaction without changing the sequence integrity of the gene. *main-2* and *mail1-1* mutant plants were transformed with the *MAIN-FLAG* or *MAIN-MYC* and *MAIL1-FLAG* or *MAIL1-MYC*

constructs, respectively, using the *Agrobacterium*-mediated floral dip method [33]. *MAIN-MYC* and *MAIL1-MYC* lines were subsequently supertransformed with the *PP7L-FLAG* construct to perform co-IP experiments. Primer sequences are described in S10 Table. The *PP7L* DNA and protein sequences used in this study are described in S12 Table.

## IP and MS analysis

Ten grams of 3-week-old seedling tissue were ground in liquid nitrogen and resuspended in 50mL ice-cold IP buffer [50mM Tris HCl pH 7.6, 150mM NaCl, 5mM MgCl$_2$, 0.1% Nonidet P-40, 10% glycerol (v/v), 0.5mM DTT, 1x Protease Inhibitor Mixture (Roche)] and centrifuged 2 times for 15 min at 4˚C at 15 350g. 400μL of M2 magnetic FLAG-beads (Sigma, M8823) were added to the supernatants, and incubated for 90 min rotating at 4˚C. M2 magnetic FLAG-beads were washed seven times in ice-cold IP buffer for 5 min rotating at 4˚C, and immunoprecipitated proteins were eluted 3 times with 150μL 3x-FLAG peptides (Sigma, F4799) for 25 min each at 25˚C. The eluted protein complexes were precipitated by trichloroacetic acid and subjected to MS analyses as previously described [13]. Peptide and protein-level false discovery rates were calculated by the DTASelect algorithm using the decoy database approach. Based on a peptide PSM level p-value filter of less than 0.01 and a requirement for at least two peptides per protein, the protein-level false discovery rate was less than 1% for all proteins detected.

## Co-IP and immunoblotting

0.5 g of 3-week-old seedling tissue were ground in liquid nitrogen, resuspended in 1.5mL ice-cold IP buffer [50mM Tris pH 7.6, 150mM NaCl, 5mM MgCl2, 0.1% Nonidet P-40, 10% glycerol, 0.5 mM DTT, 1x Protease Inhibitor Mixture (Roche)], and centrifuged 2 times for 15 min at 4˚C, 16 000g. 50μL M2 magnetic FLAG-beads (Sigma, M8823) were added to the supernatants and incubated for 2 hour rotating at 4˚C. Beads were washed 3 times in ice-cold IP buffer for 10 min rotating at 4˚C. Immunoprecipitated proteins were denatured in Laemmli buffer for 5min at 95˚C. 10μL of input and bead elution were run on 10% SDS-PAGE gels, and proteins were detected by western blotting using either Anti-FLAG M2 monoclonal antibody-peroxidase conjugate (Sigma, A8592) at a dilution of 1:10000, or c-Myc rat monoclonal antibody (Chromotek, 9E1-100) at a dilution of 1:1000 followed by goat anti-rat IgG horseradish peroxidase (Abcam, ab205720) used at a dilution of 1:20000 as secondary antibody. Western blots were developed using Substrat HRP Immobilon Western (Merck Millipore, WBKLS0500).

## RNA extraction

Total RNA was extracted from aerial parts of 3-week-old seedlings grown on soil using either RNeasy Plant Mini Kit (Qiagen, 74904) or Monarch Total RNA Miniprep Kit (NEB, T2010) according to the manufacturer's protocols.

## RNA sequencing

RNA-seq libraries were generated from 1μg of input RNA using NEBNext Ultra II Directional RNA Library Prep Kit for Illumina (NEB, E7490) according to the manufacturer's protocols. Libraries were sequenced on an Illumina HiSeq 4000 or NextSeq 550 machines. Reads were trimmed using Trimmomatic [35], and mapped to the *A. thaliana* genome (*Arabidopsis* TAIR10 genome) using HISAT2 [36]. The sequence alignment files were sorted by name and indexed using SAMtools [37]. Files were converted to BAM files and number of reads mapped onto a gene calculated using HTSeq-count [38]. Differentially expressed genes were obtained with DESeq2 [39], using a log2 fold-change $\geq 2$ (up-regulated genes) or $\leq$ -2 (down-regulated

genes) with an adjusted *p*-value of 0.01. Batch effects were modeled within the DESeq2 study design. For PCA, we removed the batch effect using limma's 'removeBatchEffect' function [40]. Heat map visualizations were realized using the heatmap2 function from the R gplots package. Boxplots were realized using boxplot function from R. Re-analyses of previously published RNA-seq datasets from *main-2* and *mail1-1* (PRJEB15202) [15] were performed as described above.

## RT-qPCR

1 µg of input RNA was converted to cDNA using GoScript Reverse Transcriptase (Promega A501C) according to the manufacturer's protocol. The final reaction was diluted 6 times with RNase free water. RT-qPCR experiments were performed with 4µL of cDNA combined to the Takyon No Rox SYBR MasterMix (Eurogentec, UF-NSMT-B0701), using a LightCycler 480 instrument (Roche). Amplification conditions were as follows: 95˚C 5 min; 45 cycles, 95˚C 15s, 60˚C 15s, 72˚C 30s; melting curves. RT-qPCR analyses used the $2^{-\Delta\Delta Ct}$ method. For each analysis, ΔCt was first calculated based on the housekeeping *RHIP1* gene Ct value [41]. ΔΔCt were then obtained by subtracting the wt ΔCt from the ΔCt of each sample. Values were represented on bar charts relative to WT. Three technical replicates were performed per biological replicate, and 3 biological replicates were used in all experiments, unless otherwise stated. Primer sequences are described in S10 Table. RT-qPCR raw data are described in S13 Table.

## DNA motif detection

The motifs for enhancer sequences (1kb upstream the TSS) were discovered using MEME (Multiple Em for Motif Elicitation). MEME represents motifs as position-dependent letter-probability matrices which describe the probability of each possible letter at each position in the pattern [42].

## Bisulfite sequencing

Genomic DNA was extracted from aerial parts of 3-week-old seedlings using Quick-DNA Plant/Seed Miniprep Kit (Zymo research, D6020) according to the manufacturer's protocol. Whole genome bisulfite sequencing (WGBS) library was prepared from 50 ng genomic DNA using NuGen Ovation Ultralow Methyl-Seq kit. Bisulfite treatment was carried out by Qiagen Epitect bisulfite kit. WGBS libraries were sequenced on an Illumina HiSeq 4000 machine. The raw reads (single end) were trimmed using Trimmomatic in order to remove adapter sequences [35]. The remaining sequences were aligned against the *A. thaliana* genome TAIR10 version using Bismark [43]. Duplicated reads were collapsed into one read. For meta-plot and boxplot visualization, we used ViewBS [44]. Boxplots were realized using boxplot function from R. DMRs (differentially methylated regions) were defined comparing methylation in wildtype with the *main-2* mutant analyzed using the R package "DMRcaller" [45]. We used "noise filter" method to compute CpG, CpHpG and CpHpH DMRs. We selected bins where the p-value was less than 0.01, the difference in methylation level was at least 40% in the CG context, 20% in the CHG context or 10% in the CHH context, with at least four cytosines; each cytosine had on average at least four reads.

## Sequence selection, multiple sequences alignments and phylogenetic reconstruction

Blast searches (blastp) were performed starting from known *A. thaliana* PMD-C and PP7/PP7L motifs on the thirty species representing the diversity of the Eudicot lineages. When

necessary tblastn searches were also used to obtain complete protein sequences. To build the phylogenetic trees, PMD-C or PP7/PP7L motifs were aligned using the multiple sequence comparison by log-expectation (MUSCLE v3.7) software [46]. Trees were reconstructed using the fast-maximum likelihood tree estimation program PHYML [47] using the LG amino acids replacement matrix [48]. Statistical support for the major clusters were obtained using the approximate likelihood-ratio test (aLRT) [49].

## Immunofluorescence and DAPI-staining

Leaves from 3-week-old plants, were fixed for 20 min rotating at 4˚C in 2% formaldehyde in Tris buffer (10 mM Tris-HCl pH 7.5, 10 mM EDTA, 100 mM NaCl), washed two times for 10 min rotating at 4˚C in cold Tris buffer and subsequently chopped in LB01 buffer (15 mM Tris-HCl pH 7.5, 2 mM EDTA, 0.5 mM spermine, 80 mM KCl, 20mM NaCl and 0.1% Triton- X-100). Nuclei were filtered through a 30 μm cell strainer cap (Sysmex, 04-0042-2316) and 5μl of the nuclei solution was diluted in 10 μl of sorting buffer (100mM Tris-HCl pH 7.5, 50 mM KCl, 2 mM MgCl2, 0.05% Tween-20 and 5% sucrose). 20μl of the nuclei dilution were spread onto a polylysine slide and air-dried for 40 min. Slides were post-fixed in 2% formaldehyde in 1X PBS for 5 min and washed 2 times with water. Slides were incubated 15 min in 1X PBS, 0.5% Triton X-100 at RT and washed 3 times with 1X PBS for 5 min. For detection, slides were incubated over night with a mouse anti-H3K9me2 monoclonal antibody (Abcam, Ab 1220) at 1:500 in 3% BSA, 0.05% Tween in 1X PBS at 4˚C in a moist chamber. After 3 washes in 1X PBS for 5 min, slides were incubated 2h with a goat anti-mouse antibody coupled to Alexa fluor 568 (Invitrogen, A11004) at 1:1000 in 3% BSA, 0.05% Tween in 1X PBS in a moist chamber. Slides were washed 1 time 5 min with 1X PBS, 1 time 10 min with 1X PBS, 1μg/mL DAPI, and 1 time 5 min with 1X PBS. DNA was counterstained with 1μg/mL DAPI in Vectashield mounting medium (Vector Laboratories). Observation and imaging were performed using a LSM 700 epifluorescence microscope (Zeiss).

## Supporting information

**S1 Fig. *MAIN* is the mutated gene responsible for *ATCOPIA28::GFP* and TE overexpression in the *ddc #16* mutant.** (A) Representative pictures of *ddc #18* (*ddc morc6-8*) and *ddc #344* (*ddc morc6-9*) mutants in comparison to *ATCOPIA28*::*GFP* WT and *ddc* control plants under UV light. Insets show plants under white light. (B) Enrichment in homozygote/heterozygote ratio of EMS over WT single nucleotide polymorphisms (SNPs), defining the linkage intervals for the populations *ddc #18* and *ddc #344*. Mb: megabase. Gray-shaded rectangles delimit the mapping intervals. (C) Location of the point mutations corresponding to the *morc6-8* and *morc6-9* alleles within the *MORC6* genomic sequence. Nucleotide and corresponding amino acid changes are indicated above the gene. Positions of the mutations are indicated relative to the transcription start site (+1). Grey boxes represent 5' and 3' UTR, blue boxes and lines represent exons and introns, respectively. (D) Enrichment in homozygote/heterozygote ratio of EMS over WT single nucleotide polymorphisms (SNPs), defining the linkage intervals for the population *ddc #16*. Gray-shaded rectangle delimits the mapping interval. (E) Location of the point mutation corresponding to the *main-3* mutant allele within the *MAIN* genomic sequence. (F) Genetic complementation analyses using the KO T-DNA insertion line *main-2*. *ddc #16* plants were crossed with *main-2* plants. F1 plants were self-crossed, and F2 plants were screened under UV light to select GFP-overexpressing plants. Western blotting using anti-GFP antibodies confirmed GFP overexpression in selected F2 plants. Coomassie staining of the large Rubisco subunit (rbcL) is used as a loading control. KDa: kilodalton. Among the selected F2 plants, the presence of *main-3* EMS and *main-2* T-DNA mutant

alleles were determined by dCAPS-PCR and PCR analyses, respectively. *DRM2* and *CMT3* genotyping were determined by PCR analyses. WT: Wild type, Ho: Homozygote mutant. He: Heterozygote. (G) Relative expression analyses of several TEs in the indicated genotypes assayed by RT-qPCR. RT-qPCR analyses were normalized using the housekeeping *RHIP1* gene, and transcript levels in the different genotypes are represented relative to WT. Error bars indicate standard deviation based on two independent biological replicates. Screening of EMS mutant populations was done on MS plates to allow for visualization of GFP-positive individuals under UV light.
(TIF)

**S2 Fig. Combining the *drm2, cmt3* and *main-3* mutations exacerbate TE silencing defects.**
(A) Principal component analysis (PCA) performed after batch correction for first two components of the sixteen samples described in RNA-seq EMS Exp1 and Exp2. (B) Relative expression analyses of *ATCOPIA28* and *HELITRONY1D* (*AT5TE35950*) in *ddc*, *main-3* and *ddc main-3* assayed by RT-qPCR. RT-qPCR analyses were normalized using the housekeeping *RHIP1* gene, and transcript levels in the different genotypes are represented relative to WT. Error bars indicate standard deviation based on three independent biological replicates. (C) Venn diagrams analysis showing the overlaps between reproducibly upregulated TEs in *ddc*, *main-3* and *ddc main-3*. Fisher's exact test statistically confirmed the significance of Venn diagram overlaps (p-value <2.2.10e-16). (D) Same as panel B for TEs defined as class I-IV TEs. Frames of RT-qPCR graphs are using the same color code as shown in panel C. (E) Venn diagrams analyses defining the overlaps between up- and downregulated genes in the different genotypes. Fisher's exact test statistically confirmed the significance of Venn diagram overlaps (p-value <2.2.10e-16). (F) Fraction of misregulated genes in *ddc*, *main-3* and *ddc main-3* located in chromosome arms or in pericentromeric regions as defined in [50]. Asterisks indicate statistically significant enrichments of misregulated genes in chromosome arms or pericentromeric regions in comparison to the genomic distributions of all *A. thaliana* genes (Chi-Square test, **: p-value≤ 0.01). Percentages of genes targeted by DNA methylation and H3K9me2 were calculated based on enrichment in heterochromatin states 8 and 9 as defined in [51]. (G) Relative expression analyses of *DRM2* and *CMT3* in *ddc*, *main-3*, *ddc main-3*, *cmt3 main-3* and *dd main-3* assayed by RT-qPCR. RT-qPCR analyses were normalized using the housekeeping *RHIP1* gene, and transcript levels in the different genotypes are represented relative to WT. Error bars indicate standard deviation based on three independent biological replicates. Screening of EMS mutant populations was done on MS plates to allow for visualization of GFP-positive individuals under UV light.
(TIF)

**S3 Fig. Identification of reproducibly misregulated loci in *main-2, mail1-1* and *main-3*.** (A) Principal component analysis (PCA) performed after batch correction for first two components of the twenty-four *main-2*, *mail1-1* and WT Col samples described in RNA-seq Exp1, Exp2 and Exp3. (B-D) Relative expression analyses of several upregulated TEs (B), upregulated genes (C), and downregulated genes (D) in *main-2*, *mail1-1* and *main-3* assayed by RT-qPCR. RT-qPCR analyses were normalized using the housekeeping *RHIP1* gene, and transcript levels in the different genotypes are represented relative to respective WT controls. Error bars indicate standard deviation based on three independent biological replicates. (E) Venn diagrams analyses representing the overlaps between misregulated loci in *main-2*, *mail1-1*, *ddc* and *ddc main-3*. Fisher's exact test statistically confirmed the significance of Venn diagram overlaps (p-value <0.005).
(TIF)

**S4 Fig. DNA methylation analyses in the *main-2* mutant.** (A-B) Boxplot analyses in two *main-2* and WT Col biological replicates showing the DNA methylation levels at genomic sites previously defined as hypo CHG differentially methylated regions (DMR) in *cmt3* (A) and hypo CHH DMR in *drm1 drm2* (B) based on [26]. p-values were calculated using a Wilcoxon test. *: p-value <5.10e-7, **: p-value <5.10e-10, ***: p-value < 2.10e-16.
(TIF)

**S5 Fig. MAIN, MAIL1 and PP7L are required for the proper expression of similar loci, and commonly downregulated genes carry the 'DOWN' DNA motif in their promoter.** (A) Principal component analysis (PCA) performed after batch correction for first two components of the thirty-two samples described in RNA-seq Exp1, Exp2 and Exp3. (B) Number of misregulated genes in the different genotypes in comparison to WT Col plants from RNA-seq Exp3 (four biological replicates, S3 and S6 Tables). (C) Number of upregulated TEs in *pp7l-2* and *mail1-1 pp7l-2*, and classified by TE superfamily. (D) Fraction of misregulated loci in *pp7l-2* and *mail1-1 pp7l-2* located in chromosome arms or in pericentromeric regions as defined in [50]. Asterisks indicate statistically significant enrichments of downregulated genes, upregulated genes and TEs in chromosome arms and pericentromeric regions, respectively, in comparison to the genomic distributions of all *A. thaliana* genes and TEs (Chi-Square test, *: p-value≤ 0.05, **: p-value≤ 0.01, n.s: not significant). Percentages of genes targeted by DNA methylation and H3K9me2 were calculated based on enrichment in heterochromatin states 8 and 9 as defined in [51]. (E) Identification and proportions of the 'DOWN' DNA motif among the promoters of downregulated genes and all *Arabidopsis* genes using the MEME software. Promoter regions are defined as 1kb upstream of ATG. The list of all *Arabidopsis* genes used to determine genomic distributions is based on the TAIR file: TAIR10_upstream_1000_translation_start_20101028. RNA-seq threshold: log2≥2, or log2≤-2; p-adj< 0.01.
(TIF)

**S6 Fig. Full size images of Western blot panels described in Fig 1C, Fig 5B and Fig 5G–5I.**
(TIF)

**S1 Table. Lists of differentially expressed loci in *ddc, main-3* and *ddc main-3*.**
(XLSX)

**S2 Table. Lists of differentially expressed loci in *main-2* and *mail1-1*.**
(XLSX)

**S3 Table. Lists of loci commonly misregulated in *main-2, mail1-1* and *main-3*.**
(XLSX)

**S4 Table. Lists of differentially expressed loci in *pp7l-2* and *mail1-1 pp7l-2*.**
(XLSX)

**S5 Table. Lists of loci commonly misregulated in *main-2, mail1-1, pp7l-2* and *mail1-1 pp7l-2*.**
(XLSX)

**S6 Table. Lists of loci commonly misregulated in all mutant backgrounds (except *ddc*) analyzed in this study.**
(XLSX)

**S7 Table. Lists of commonly downregulated genes displaying the "DOWN" motif in their promoter and random test analyses.**
(XLSX)

**S8 Table. List of species used to construct the two trees of Fig 8, their codes and the presence/absence of the different PMD-C and PP7 motifs.**
(XLSX)

**S9 Table.** (A) PMD-C and (B) PP7/PP7L motifs used to construct the two phylogenetic trees of Fig 8.
(XLSX)

**S10 Table. List of primers used in this study.**
(XLSX)

**S11 Table. Next Generation Sequencing (NGS) mapping and coverage statistics.**
(XLSX)

**S12 Table. Genomic DNA and protein sequences of *PP7L (AT5G10900)* used in this study.**
(XLSX)

**S13 Table. Raw data of RT-qPCR experiments described in this study.**
(XLSX)

## Acknowledgments

The authors want to thank Thierry Lagrange, Frederic Pontvianne and other team members for fruitful discussions, and all the LGDP platform members for their outstanding technical assistance and plant care. The authors also thank Remy Merret and Michele Laudie for their precious help as part of the UPVD Bio-environment facility.

## Author Contributions

**Conceptualization:** Guillaume Moissiard.

**Data curation:** Nathalie Picault, Yasaman Jami-Alahmadi, Suhua Feng, Etienne Bucher, James Wohlschlegel.

**Funding acquisition:** Guillaume Moissiard.

**Investigation:** Melody Nicolau, Nathalie Picault, Julie Descombin, Yasaman Jami-Alahmadi, Suhua Feng, Etienne Bucher, Jean-Marc Deragon, James Wohlschlegel, Guillaume Moissiard.

**Resources:** Steven E. Jacobsen.

**Supervision:** Guillaume Moissiard.

**Validation:** Melody Nicolau, Julie Descombin.

**Visualization:** Melody Nicolau, Nathalie Picault, Etienne Bucher, Jean-Marc Deragon.

**Writing – original draft:** Jean-Marc Deragon, Guillaume Moissiard.

**Writing – review & editing:** Melody Nicolau, Nathalie Picault, Julie Descombin, Steven E. Jacobsen, Jean-Marc Deragon, Guillaume Moissiard.

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
