## [Decision Letter · Decision Letter 0]

16 Aug 2019

Dear Dr Moissiard,

Thank you very much for submitting your Research Article entitled 'The plant mobile domain proteins MAIN and MAIL1 interact with the phosphatase PP7L to regulate gene expression and silence transposable elements in Arabidopsis thaliana.' to PLOS Genetics. Your manuscript was fully evaluated at the editorial level and by independent peer reviewers. The reviewers appreciated the attention to an important problem, but raised some substantial concerns about the current manuscript. In particular, the reviewers asked to substantiate the relationship between MAIN/MAIL1 and PP7L by adding detailed information on changes of the transcriptome and methylome and by adding information on double mutants of main/mail1 and pp7l. I agree that adding additional information would strengthen this manuscript. The reviewers made several suggestions to improve the transcriptome and methylome analyses that the authors could follow. Based on the methods I had the impression the authors only analyzed TE genes for expression changes rather than all TEs, which may explain the low number of detected changes for TEs. I would also ask the authors to complement the metagene plots with boxplots to allow a statistical assessment of possible changes of DNA methylation.

Based on the reviews, we will not be able to accept this version of the manuscript, but we would be willing to review again a much-revised version. We cannot, of course, promise publication at that time.

If you decide to revise the manuscript for further consideration at PLOS Genetics, please aim to resubmit within the next 60 days, unless it will take extra time to address the concerns of the reviewers, in which case we would appreciate an expected resubmission date by email to plosgenetics@plos.org.

[LINK]

We are sorry that we cannot be more positive about your manuscript at this stage. Please do not hesitate to contact us if you have any concerns or questions.

Yours sincerely,

Claudia Köhler

Associate Editor

PLOS Genetics

Wendy Bickmore

Section Editor: Epigenetics

PLOS Genetics

Reviewer's Responses to Questions

**Comments to the Authors:**

Reviewer #1: see attachment

Reviewer #2: This is a review of the manuscript entitled “The plant mobile domain proteins MAIN and MAIL1 interact with the phosphatase PP7L to regulate gene expression and silence transposable elements in Arabidopsis thaliana” submitted to PLoS Genetics. In this manuscript a forward genetic screen is performed to identify factors responsible for repressing a TE-reporter transgene in an already sensitized DNA methylation-deficient background. The screen hits upon MAIN, a transposable element (TE)-derived domesticated protein that has been described before, but the molecular function of which is unknown. The strength of the manuscript is the analysis of TE activation in Figure 1 and the IP-Mass Spec analysis in Figure 2. The first two figures, along with their corresponding supplemental data, are very strong. Figure 3 and 4 are the weakness of the manuscript, as a more convincing analysis of dataset overlap needs to be done for Figure 3, and Figure 4 is highly descriptive, and ends the manuscript on a low note. Overall, it is my belief that upon revision this work could be suitable for publication in PLoS Genetics.

Major comments

1. The analysis of Figure 3A-C is not convincing to the reader. Rather than just explaining how many genes or TEs are shared between datasets, the others should perform a statistical analysis of observed compared to expected overlap based on the size of these datasets, and show p-values for how statistically enriched the overlap is. The same for the Venn diagram in Figure 1E. For example, there are so many genes and TEs in the genome, that the numbers of overlap in Figure 3A-C may simply be due to the false discovery rate (especially the very low numbers of Figure 3A). Besides this observed vs. expected analysis, to convince the readers that MAIN, MAIL1 and PP7L have similar sets of mis-expressed genes and TEs, the authors should perform a principle component analysis that shows that the expression patterns of these mutant genotypes cluster away from wild-type, and more importantly they cluster together.

2. Figure S1F shows that crossing the TE-reporter transgene to the T-DNA insertion of MAIN gives the same effect as the point mutation from their screen. However, I am not convinced by the GFP fluorescence images nor the Western blot. The main mutant combined with ddc looks just like the starting ddc to me. This is an important point, and convincing data needs to be show that MAIN is the correct gene causing the effect identified in the screen. Perhaps GFP levels could be quantified in S1F?

3. I’m amazed that MAIN and MAIL1 coIP with PP7L, which just happens to be very close to the MAIL3 gene, in which two similar proteins to MAIN/MAIL1 and PP7L have been fused together. Please add discussion of this coincidence, and how it could have come to be that two proteins that work together could become fused.

4. The end of the Results section becomes highly descriptive in regards to the evolutionary analysis. My suggestion is to either perform an experiment (bioinformatic or wet-bench) in this section, or reduce this analysis / description.

Minor comments, but still should be fixed

1. The Introduction goes into background research on MORC and MOM1 proteins that is not necessary for the reader, and I feel that the purpose of mentioning them is to self-reference the authors’ previous work. Extraneous information and references should be removed.

2. Figures S1-S3 have a lot of good and important information in them. I suggest moving the essential pieces of data out of the Supplemental section and into the main figures.

3. More raw data should be added to Figure 2A (the IP-MS table). For example: % of the detected protein covered by the spectra, p-value of detection, number of unique spectra for that protein.

4. Figure S3A-C should be in the main figures. For Figure3B-C please show comparable data for MAIN and MAIL1, and see major point #1 above for a principle component analysis that could convince the reader that these mutants have similar expression patterns.

Reviewer #3: The manuscript of Nicolau and colleagues investigates the function of two Arabidopsis plant mobile domain (PMD) genes MAIN and MAIL1 in the regulation of transcription. In a forward genetic screen, the authors identify a point mutation in MAIN leading to the reactivation of a silent GFP reporter. RNA-sequencing shows that loss of MAIN leads to derepression of transposable elements (TEs) in redundant and synergistic manners to DNA methylation. Immunoprecipitation followed by mass spectrometry (IP-MS) shows that MAIN and its homolog MAIL1 are in a protein complex with the putative phosphatase PP7L. Comparison between different publically available gene expression datasets defines a common set of TEs and protein-coding genes mis-regulated in main, mail1 and pp7l mutants. Motif analyses identify a putative cis regulatory element (CRE) amongst mis-regulated genes and phylogenetic work suggests a neo-association process between PMD and PP7 domains.

The role of MAIN and MAIL1 in TE silencing is of general interest for the plant epigenetic community and the identification of PP7L as part of the MAIN/MAIL protein complex is the novelty of this work. The originality and hard work involved in performing a genetic screen has to be noted. However, many of the manuscript findings have already been published (Ikeda et al., Nature Communication 2017; DOI: 10.1038/ncomms15122). This encompasses the identification of MAIN and MAIL1 as transcriptional repressors of TEs, independently of DNA methylation. Unfortunately, the authors do not explore in more details the role of PP7L in gene expression and its interplay with MAIN/MAIL1, which are the major discoveries of this manuscript. The analysis of high-order mutants between main/mail and pp7l mutants would shed light on how these partners function. Also, there are multiple weaknesses in the methodology that need to be improved before publishing. With additional data deciphering the function of PP7L in transcriptional regulation, this work could be considered for publication in Plos Genetics.

In summary, the research and approaches are interesting but further experiments and analyses are required. I therefore positively encourage the authors to do revisions and resubmit this manuscript.

Major reviews and comments:

In general:

• Explanations in main text are often brief and sparse, thus making it difficult to understand the manuscript. Examples: genetic complementation and crosses, co-immunoprecipitations.

Genetic complementation Fig S1F:

• Cross of ddc#16 with main-2: In F2, there is no information on the genotype of the MAIN gene. Thus complementation cannot be assessed. I believe that the authors forgot to mention that all the selected GFP positive plants were either homozygous for main-2 or main-3, which solves this issue.

• Is ATCOPIA::GFP reactivated in main-2 single mutant? Are F1 plants GFP positive? Homozygosity of CMT does not seem to be required for GFP reactivation in F2.

• Complementation can also be tested by transforming ddc#16 with the available flag-tagged MAIN construct.

RNA-seq ddc main-3:

• This genome-wide information could be used to illustrate in more details the synergistic or redundant relationship between MAIN and DRM2/CMT3 of all individual up-regulated TEs. Figure 1D could for example be presented as heat map. For example, are the TEs up-regulated in the double mutant already partially reactivated in the single mutants?

• What are the specific contributions of CHG and CHH methylation to TE reactivation in addition to loss of MAIN? The non-requirement of CMT3 homozygosity for GFP reactivation in F2 plants (Figure S1F) suggests that CHH, but not CHG methylation is the major silencing pathway redundant with MAIN. Looking at TE expression by RT-qPCR and GFP reactivation in double mutants, such as drm2/main-3 or cmt3/main-3, segregating from the complementation cross would be informative.

• Are there protein-coding genes changing expression in these genotypes? Are these genes overlapping with the analysis performed in figure 3 with the publically available main-2 RNA-seq data?

• The growth conditions (soil or plates) of the 3 weeks old seedlings remain unclear.

• The link to the deposition of the data is not valid.

Interactions MAIN/MAIL1/PP7L:

• Protein ladder sizes are missing on western blots in figure 2B-D. It would be important to add the images of the entire blots with all the protein sizes in the supplements.

• The full list of identified proteins is missing. Were there additional interesting interaction partners identified with MAIN and MAIL?

• From the presented data, it cannot be ruled out that an additional protein is acting as bridge between MAIN/MAIL and PP7L. IP-MS of flag-tagged PP7L would strengthen the proposed protein complex. Also, has a bzip transcription factor been identified? In the discussion, the authors suggest that the MAIN/MAIL/PP7L complex recruit a transcription factor that could recognize the identified CRE motif.

• The data submitted to the MassiVE database cannot be accessed.

Transcriptional profiling of main-2, mail1-1 and pp7l-2:

• The plant growth stages and conditions between the re-analysed RNA-seq experiments are very different thus not providing ideal datasets to compare transcriptional responses between these genotypes. Main-2 and mail1-1 mutants were grown 18 days old on soil while the pp7l mutant was grown for 4 days on sterile plates supplemented with 1% sucrose.

• Validation RT-qPCRs were performed on 3 weeks old seedlings (grown on plates or on soil?) however the mail1-1 mutant is missing.

• To understand the functional relationship between MAIN/MAIL and PP7L, it would be important to investigate transcriptional responses in double and triple mutants. This would reveal if PP7L functions exclusively with the MAIN/MAIL pathway or if PP7L has an additional, MAIN/MAIL-independent role in transcription.

Motif analysis:

• For the identification of specific CRE motifs, have the authors also tested if randomly selected genes do not identify such motifs? Similarly, do genes only up-/down-regulated in single mutants have such motifs?

• Is the CRE motif identified in up-regulated genes also found upstream of up-regulated TEs? At the genome-wide scale, how many genes do possess such motives in their promoters?

• Are MAIN, MAIL or PP7L binding the identified motifs? This could for example be tested by chromatin immunoprecipitation experiments in vivo or in vitro using EMSA or SELEX.

DNA methylation and chromatin compaction:

• To further show that loss of MAIN does not impact DNA methylation, DNA methylation levels should be plotted for all protein-coding genes, all TEs, and more specifically for sites that lost CHH or CHG methylation in drm2 and cmt3 mutants respectively.

• Have the authors looked if there were subtle changes? BS-seq data allows calling differentially methylated regions (DMRs). Are there any DMRs in the main-2 mutant?

• Main and mail1 mutants were shown to display chromocenters decondensation. Is this also the case in pp7l-2? This would support a direct role for PP7L in chromatin compaction and strengthen the relationship between MAIN/MAIL1 and PP7L.

Co-evolution analysis:

• The resolution of figure 4 is low and cannot be read.

**Have all data underlying the figures and results presented in the manuscript been provided?**

Reviewer #1: Yes

Reviewer #2: Yes

Reviewer #3: No: The links for the deposition of the RNA-seq and IP-MS data are not valid.

PLOS authors have the option to publish the peer review history of their article (what does this mean?). If published, this will include your full peer review and any attached files.

Reviewer #1: No

Reviewer #2: No

Reviewer #3: No

---

## [Decision Letter · Decision Letter 1]

22 Dec 2019

Dear Dr Moissiard,

Thank you very much for submitting your Research Article entitled 'The plant mobile domain proteins MAIN and MAIL1 interact with the phosphatase PP7L to regulate gene expression and silence transposable elements in Arabidopsis thaliana.' to PLOS Genetics. Your manuscript was fully evaluated at the editorial level and by two independent peer reviewers. The reviewers acknowledged that the revisions strongly improved the manuscript, but one of the reviewers identified several aspects that should be improved. I agree with the reviewer that the transcriptome analyses requires revisions; the highly stringent cutoffs applied together with the fact that several experiments were independently analyzed and only overlaps further considered, limits the number of identified loci, which may impact on the conclusions. Based on the reviews, we will not be able to accept this version of the manuscript, but we would be willing to review again a revised version addressing those concerns.

Your revisions should address the specific points made by the reviewer. We will also require a detailed list of your responses to the review comments and a description of the changes you have made in the manuscript.

If you decide to revise the manuscript for further consideration at PLOS Genetics, please aim to resubmit within the next 60 days, unless it will take extra time to address the concerns of the reviewers, in which case we would appreciate an expected resubmission date by email to plosgenetics@plos.org.

[LINK]

We are sorry that we cannot be more positive about your manuscript at this stage. Please do not hesitate to contact us if you have any concerns or questions.

Yours sincerely,

Claudia Köhler

Associate Editor

PLOS Genetics

Wendy Bickmore

Section Editor: Epigenetics

PLOS Genetics

Reviewer's Responses to Questions

**Comments to the Authors:**

Reviewer #1: see attachment

Reviewer #3: Thank you very much for all the improvements and the addition of new interesting data to this manuscript.

**Have all data underlying the figures and results presented in the manuscript been provided?**

Reviewer #1: Yes

Reviewer #3: Yes

PLOS authors have the option to publish the peer review history of their article (what does this mean?). If published, this will include your full peer review and any attached files.

Reviewer #1: No

Reviewer #3: No

---

## [Decision Letter · Decision Letter 2]

14 Feb 2020

Dear Dr Moissiard,

Thank you very much for submitting your Research Article entitled 'The plant mobile domain proteins MAIN and MAIL1 interact with the phosphatase PP7L to regulate gene expression and silence transposable elements in Arabidopsis thaliana.' to PLOS Genetics. Your manuscript was fully evaluated at the editorial level and by independent peer reviewers. One of the reviewers identified two minor issues to be addressed before the manuscript can be finally accepted.

We therefore ask you to modify the manuscript according to the review recommendations before we can consider your manuscript for acceptance.

[LINK]

Yours sincerely,

Claudia Köhler

Associate Editor

PLOS Genetics

Wendy Bickmore

Section Editor: Epigenetics

PLOS Genetics

Reviewer's Responses to Questions

**Comments to the Authors:**

Reviewer #1: In their revised manuscript Nicolau et al. addressed all my previous concerns and now present a much more compelling set of transcriptomics data. I have no addition major issues and the minor issues listed should not warrant additional review.

Minor Comments

1. Fig 1C, please add the raw image of the gel to the figure with the rest of the gels.

2. Fig 5B, the gel is labeled MAIN and MAIL1-Flag, but the gels are Flag and Myc. Please relabel for clarity and also add the raw images of these gels to the figure with the rest of the gels.

**Have all data underlying the figures and results presented in the manuscript been provided?**

Reviewer #1: Yes

PLOS authors have the option to publish the peer review history of their article (what does this mean?). If published, this will include your full peer review and any attached files.

Reviewer #1: No

---

## [Editor Report · Decision Letter 3]

28 Feb 2020

Dear Dr Moissiard,

We are pleased to inform you that your manuscript entitled "The plant mobile domain proteins MAIN and MAIL1 interact with the phosphatase PP7L to regulate gene expression and silence transposable elements in Arabidopsis thaliana." has been editorially accepted for publication in PLOS Genetics. Congratulations!

Yours sincerely,

Claudia Köhler

Associate Editor

PLOS Genetics

Wendy Bickmore

Section Editor: Epigenetics

PLOS Genetics

Comments from the reviewers (if applicable):

**Data Deposition**

http://datadryad.org/submit?journalID=pgenetics&manu=PGENETICS-D-19-01185R3

**Press Queries**

---

## [Editor Report · Acceptance letter]

26 Mar 2020

PGENETICS-D-19-01185R3 

The plant mobile domain proteins MAIN and MAIL1 interact with the phosphatase PP7L to regulate gene expression and silence transposable elements in Arabidopsis thaliana. 

Dear Dr Moissiard, 

We are pleased to inform you that your manuscript entitled "The plant mobile domain proteins MAIN and MAIL1 interact with the phosphatase PP7L to regulate gene expression and silence transposable elements in Arabidopsis thaliana." has been formally accepted for publication in PLOS Genetics! Your manuscript is now with our production department and you will be notified of the publication date in due course.

With kind regards,

Kaitlin Butler

PLOS Genetics

On behalf of:
